



# Dating the ice of Gauligletscher, Switzerland, based on surface radionuclide contamination and ice flow modeling

Guillaume Jouvet[1,2,†], Stefan Röllin[3], Hans Sahli[3], José Corcho[3], Lars Gnägi[4,5], Loris Compagno[6,7], Dominik Sidler[5,8], Margit Schwikowski[4,9], Andreas Bauder[6], and Martin Funk[6]

[1]Autonomous Systems Laboratory, ETH Zurich, Switzerland
[2]Department of Geography, University of Zurich, Switzerland
[†]formerly at the Laboratory of Hydraulics, Hydrology and Glaciology, ETH Zurich, Switzerland
[3]Spiez Laboratory, Federal Office for Civil Protection, Switzerland
[4]Department of Chemistry and Biochemistry, University of Bern, Switzerland
[5]NBC Defence Laboratory 1, Swiss Armed Forces
[6]Laboratory of Hydraulics, Hydrology and Glaciology, ETH Zurich, Switzerland
[7]Swiss Federal Institute for Forest, Snow and Landscape Research, Switzerland
[8]Max Planck Institute for the Structure and Dynamics of Matter, Hamburg, Germany
[9]Paul Scherrer Institute, Switzerland

**Correspondence:** guillaume.jouvet@mavt.ethz.ch

**Abstract.** In the 1950s and '60s, specific radionuclides were released into the atmosphere as a result of nuclear weapons testing. This radioactive fallout left its signature on the accumulated layers of glaciers worldwide, thus providing a tracer for ice particles traveling within the gravitational ice flow and being released into the ablation zone. For surface ice dating purposes, we analyze here the activity of $^{239}$Pu, $^{240}$Pu and $^{236}$U radionuclides derived from more than 200 ice samples collected along

five flowlines at the surface of Gauligletscher, Switzerland. It was found that contaminations appear band-wise along most of the sampled lines, revealing a V-shaped profile consistent with the ice flow field already observed. Similarities to activities found in ice cores permit the isochronal lines at the glacier from 1960 and 1963 to be identified. Hence this information is used to fine-tune an ice flow/mass balance model, and to accurately map the age of the entire glacier ice. Our results indicate the strong potential for combining radionuclide contamination and ice flow modeling in two different ways. First, such tracers

provide unique information on the long-term ice motion of the entire glacier (and not only at its surface), and on the long-term mass balance, and therefore offer an extremely valuable data tool for calibrating ice flows within a model. Second, the dating of surface ice is highly relevant when conducting "horizontal ice core sampling", i.e., when taking chronological samples of surface ice from the distant past, without having to perform expensive and logistically complex deep ice-core drilling. In conclusion, our results show that an airplane which crash-landed on the Gauligletscher in 1946 will likely soon be released

from the ice close to the place where pieces have emerged in recent years, thus permitting the prognosis given in an earlier model to be revised considerably.



## 1   Introduction

Climate change is expected to bring about the massive retreat of glaciers worldwide (e.g. Marzeion et al., 2018) with important consequences for hydropower production (Schaefli et al., 2019), the tourism economy (Wiesmann et al., 2005), or in terms
of sea level rise (Mengel et al., 2018). It is therefore of major interest to model the future extents of glaciers in order to anticipate the projected change as accurately as possible. To meet the need for prediction tools, numerical models combining ice flow and surface mass balance have been developed at different levels of complexity, from flowline models based on mechanical approximations (e.g. Wallinga and van de Wal, 1998) to three-dimensional models based on full Stokes equations (e.g. Gagliardini et al., 2013). A common difficulty in setting up sophisticated models is the lack of data to simultaneously
constrain the key parameters that control the mass balance, the ice deformation, and the basal conditions; this can be a source of major model uncertainties (Gillet-Chaulet et al., 2012). For instance, both mass balance measurements, and ice velocities observed at the glacier surface, are generally available only for the ablation area and for the recent past. As a result models rely essentially on sparse data that ignore the dynamics of the upper glacier part, and rarely integrate long time series that account for the decadal-scale variability of the ice movements (glaciers were in general thicker and then faster in the past). Another
important limitation is that models rarely integrate data on the basal motion at the bedrock, for reasons of inaccessibility (Vincent and Moreau, 2016).

One way to tackle this issue is to include Lagrangian information on ice particles traveling from the accumulation to the ablation zone whenever we have proxy data to track their trajectories. The surface ice is expected to be young near the equilibrium line altitude and to get older in the direction of the glacier snout (Fig. 1), while the trajectories followed by the ice tend
to become longer and slower. Therefore, data on the age of the ice released in the ablation area is extremely valuable because it integrates information over a long time period, and as it indicates the displacement of deep ice particles which might be influenced by basal conditions. To our knowledge, Lagrangian data – which includes whatever datable tracer is found in the ablation area – has never been used to calibrate a dynamic glacier model of an Alpine glacier, as such data is rare. Reeh et al. (2002) used a flowline model to determine the trajectory of ice particles within the Greenland ice sheet and to date surface ice
near the margin, by comparing the $\delta^{18}$O values of samples to ice core reference values. Jouvet and Funk (2014) used a three-dimensional ice flow model to reconstruct the trajectory of the corpses of three ill-fated mountaineers missing since 1926, and which emerged from the surface of Aletschgletscher in 2012. Using the mountaineer's bodies as a tracer for the age of adjacent ice particles, it was possible to validate the model independently. Following the same methodology, Compagno et al. (2019) modeled the dynamics and the evolution of Gauligletscher and used the model results to reconstruct the space-time trajectory
of the Dakota airplane, that crash-landed on the highest part of the glacier in 1946, in order to estimate its present position and when and where it will emerge at the surface. Other ice flow models have been applied to compute the age of ice in a prognostic way, e.g. to determine the location of the oldest ice in Antarctica (Sutter et al., 2019) or to identify isochrone geometry of ice sheets (Hindmarsh et al., 2009). More locally, models accounting for the compressibility of firn at the accumulation area of glaciers have been used to infer the depth-age relationship of ice cores (e.g. Lüthi and Funk, 2000; Schwerzmann, 2006; Winski
et al., 2017). Recently, Licciulli et al. (2019) improved this approach by combining to an ice flow model.



Existing ice dating methods have been designed mainly for ice cores extracted in accumulation zones with minor ice dynamics to permit the age-depth relationship to be more easily understood. For instance, Eichler et al. (2000) combined multiple methods for dating an ice core extracted at the Grenzgletscher, Switzerland, allowing a yearly accuracy to be achieved. Their methodology includes tracers based on radioactive isotope $^{210}$Pb (Gäggeler et al., 1983) and seasonally varying signals such as the concentrations of $NH_4^+$ and the isotopic ratio $\delta^{18}$O, as well as using debris embedded within the ice originating from well-known events such as Saharan dust falls, atmospheric Nuclear Weapon Tests (NWT) and the reactor accident in Chernobyl (Haeberli et al., 1988). Among them, the NWT fallouts related to the 1954-1967 period have been well detected in many ice cores (e.g. Olivier et al., 2004; Gabrieli et al., 2011; Wang et al., 2017) and other archives (e.g., in lake sediments, Röllin et al., 2014) by analyzing the activity of $^{239}$Pu, $^{240}$Pu and $^{236}$U radionuclides. No similar dating has been performed at the surface of the ablation area of a glacier, and the question whether the aforementioned proxies are identifiable after experiencing intense ice deformation and travelling into temperate ice with close proximity to percolation water remains opened. To our knowledge, a single unpublished work by Uwe Morgenstern describes the tritium-based dating of surface ice of mountain glaciers. Recently, (Gäggeler et al., 2020) used $^{210}$Pb activity concentration decays of surface ice samples collected in the ablation zone of Aletschgletscher to determine relative ages between samples and deduce ice flow rates. Nevertheless, knowing the absolute age of glacier surface ice would be highly useful for conducting "horizontal ice cores", i.e., collecting and analyzing old ice (the proxies that have not been altered by ice thermo-mechanical changes) without having to perform expensive and logistically complex deep ice core drilling (Reeh et al., 2002; Baggenstos et al., 2018). The main challenge when doing so is to find undisturbed isochrones, and to sample across them. Therefore, the efficiency of the method is strongly dependent on our capacity to map the age of the surface ice in order to derive an accurate distance-age relationship – a field of research largely unexplored by ice flow modelers.

In this paper, we combine for the first time ice flow modeling and ice dating based on radionuclide contamination to derive an accurate map of the age of ice at Gauligletscher. First we use the results of a former model (Compagno et al., 2019) in order to identify the most likely region of the ablation area where ice from the early 1950s and the late 1960s might exist, as this zone being expected to be contaminated by radioisotopes due to fallouts after atmospheric NWT. Then we analyze $^{239}$Pu, $^{240}$Pu and $^{236}$U activities from more than 200 ice samples, which have been collected throughout the region identified by the model, to date the ice in a direct way. We take advantage of this new data to refine our ice flow model, to date the entire glacier ice, and to update our prediction of the location of the Dakota airplane (Compagno et al., 2019).

This paper is organised as follows: First, we give a short description of Gauligletscher and the data available to run the model. Then we present the methods for dating the ice based on ice flow modeling and radionuclide contamination. Following that, we present our results and discuss the implications of the new data set for refining the model as well as the implications for the current positions of the wreckage of the Dakota airplane.



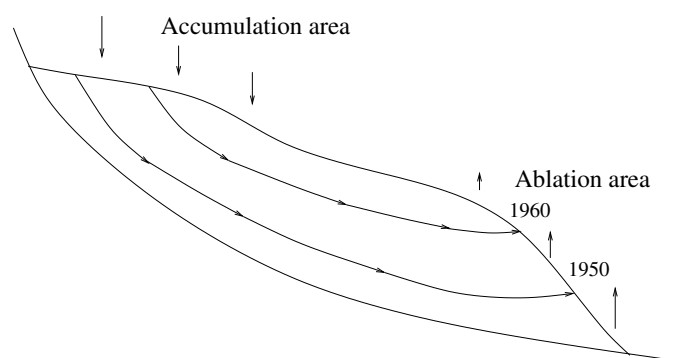

**Figure 1.** Conceptual representation of ice particles traveling within a glacier ice flow from the accumulation to the ablation area. The oldest ice released is the one that has the longest and slowest trajectories such as the ice found at the glacier snout. Two examples are given based on ice originating from 1960 and 1950.

## 2 Study site

Gauligletscher (Fig. 2) is a ≈6 km-long glacier covering an area of ≈11 km$^2$ located in the Bernese Alps, Switzerland (data from 2010). The glacier has retreated significantly since the start of the monitoring in the 1950s. According to (GLAMOS, 85 2018), its tongue lost more than one kilometer between 1958 and 2018.

Generally speaking, Gauligletscher has entered our awareness since November 1946, when a Douglas C-53 Dakota airplane crash-landed on its upper accumulation area (Fig. 2). After operations to rescue passengers shortly after and to recover the most valuable pieces in summer 1947, the Dakota main body was left on site, buried gradually under seasonal snow accumulation and transported downstream within the ice flow. Although the plane's pieces were released into the ablation area of the glacier 90 and found between 2012 and 2018, the main body of the Dakota still remains embedded in the ice.

In a previous study, Compagno et al. (2019) modeled the dynamics and the evolution of Gauligletscher and used the model results to reconstruct the space-time trajectory of the airplane to estimate its present position and when and where it will reappear at the surface. The results suggest that the main body of the airplane will be released between approximately 2027 and 2035, 1 km upstream of the parts that emerged between 2012 and 2018. To explain their separation from the main body, 95 Compagno et al. (2019) suggested that the recently found pieces of the Dakota might have been removed from the original airplane location and moved downglacier before being abandoned in 1947.

## 3 Methods

We now describe in order the methods we used to model the age of ice using both ice flow modeling and radionuclide contam- 100 ination.





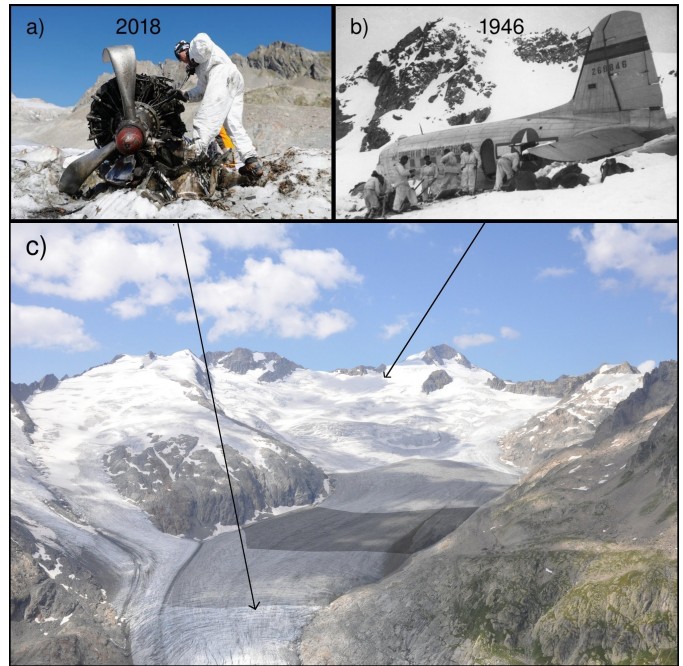

**Figure 2.** a) The engine released at the surface of the glacier and found in 2018 (Foto VBS, source: www.vtg.admin.ch), b) the Dakota aircraft during the rescue operation in 1946 (Foto MHMLW, source: www.vtg.admin.ch), c) Aerial view of Gauligletscher taken in summer 2019 (Photo: Lars Gnägi). The light and dark shaded areas indicate the low and high resolution sampled zones, respectively.

## 3.1 Glacier model

Prior to sampling, we used the results of Compagno et al. (2019), who modeled the ice flow field and the time evolution of the surface of Gauligletscher from 1947 forward in time, to estimate the region of the glacier where it could be expected that ice from the early 1950s and the late 1960s would be released at the surface. After sampling and data analysis (Section 4), we

improved the model and performed a range of new model runs, which are presented in this study.

Our model uses the Elmer/Ice finite-element software (Gagliardini et al., 2013) to iteratively compute the ice flow velocity field and the mass balance, and therefore to simulate the time evolution of Gauligletscher, while providing the initial glacier surface DEM, the bedrock DEM, model parameters and meteorological data. In the next section, we describe the data, the ice flow, the mass balance and the age of ice models. Then we describe our strategy for calibrating the model parameters.

### 3.1.1 Data

Here we used a wide range of data to run our model, including Digital Elevation Models (DEMs) from 1947 and 2010, several ground penetrating radar profiles (Rutishauser et al., 2016) with a resulting bedrock elevation model (Farinotti et al., 2009) (Fig. 3b), recent satellite Sentinel-2 orthoimages to derive a surface ice flow field (Fig. 3a and Appendix A), as well as





meteorological time series of weather stations located nearby. For this new study, we also used the 1980 DEM, and an update

of observed velocities based on the 2015-2019 observations.

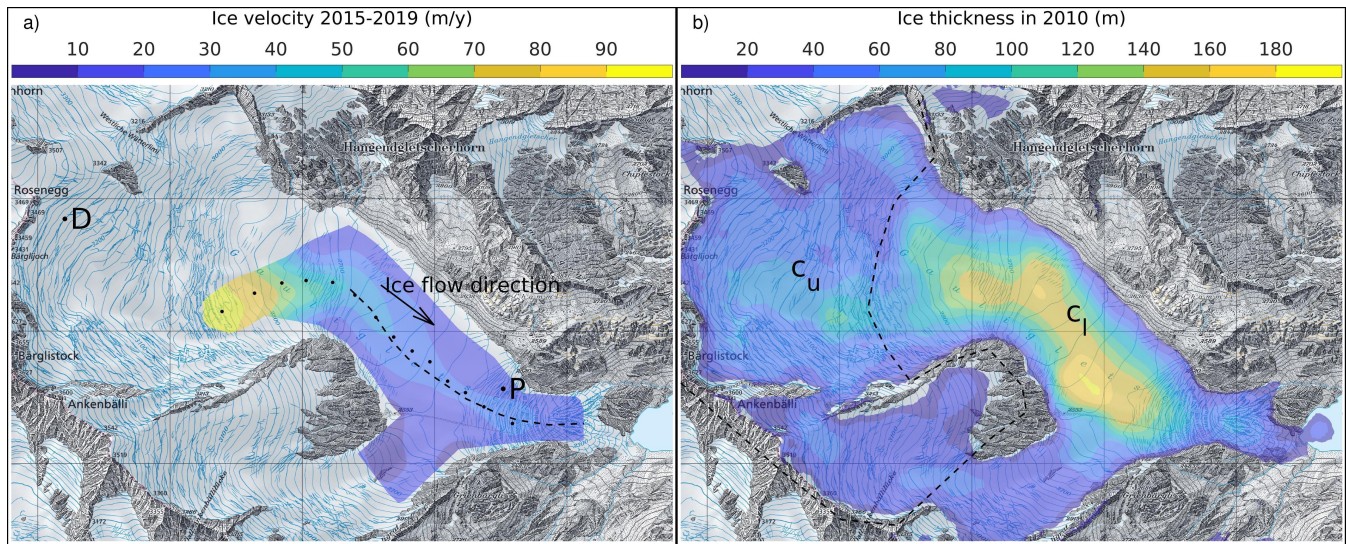

**Figure 3.** Topographic map of Gauligletscher showing a) the observed surface velocities over the period 2015-2019 derived by remote sensing (Appendix A) and b) ice thickness distribution in 2010. On Panel a), the dashed line shows the line of maximal observed velocities along each lateral transect, while the dots indicate the position where the observed modeled RMSE is computed. The location where the Dakota airplane crash-landed in 1946, and the place where pieces of the plane were released in 2018 at the surface are indicated by the letters 'D' and 'P', respectively. On Panel b), the dashed line delimits the lower and upper glacier areas where sliding is parametrized with $c_l$ and $c_u$, respectively.

### 3.1.2 Ice flow

The motion of ice is described by the full Stokes equations (Greve and Blatter, 2009) based on Glen's law, with Glen's exponent $n = 3$ and constant rate factor $A$, with ice assumed to be isothermal and temperate. Here we tested values $A = 60, 100, 150$ MPa$^{-3}$ a$^{-1}$ to assess the effect of various ice flow regimes from high shearing dominant (high $A$) to sliding dominant (low $A$).

Note that rate factors $A$ used to model similar Alpine glaciers are usually at the low end of this range, i.e., 60 to 100 MPa$^{-3}$ a$^{-1}$ (e.g. Gudmundsson, 1999; Jouvet et al., 2011) around the value $A = 78$ MPa$^{-3}$ a$^{-1}$ recommended for temperate ice by Cuffey and Paterson (2010). As a boundary condition along the bedrock, we used the following non-linear sliding law – known as Weertman's law (Weertman, 1957):

$$\sigma_{nt_i} + cu_b^m u_{t_i}, \qquad i = 1, 2, \tag{1}$$

where $\sigma_{nt_i} = \mathbf{t}_i \cdot (\sigma \mathbf{n})$, $u_b$ is the norm of the basal velocity, and $u_{t_i} = \mathbf{u} \cdot \mathbf{t}_i$ are the basal shear stresses and basal velocities, respectively, defined in terms of tangent vectors $\mathbf{t}_i$ and the normal outward-pointing unit vector $\mathbf{n}$ to the bedrock, while



$m = 1/3$ is a constant parameter and $c$ is a sliding coefficient. In our model, $c$ takes two values: $c_l$ and $c_u$ in the lower ('l') and upper ('u') parts of the glacier as defined in Fig. 3b, with a smooth transition between the two zones (not shown). The reason for using two values is that observed surface velocities used to adjust $c$ are available only in the lower part of the glacier, and thus suitable to tune $c_l$ but not $c_u$. Additionally, the dynamics of the lower part is expected to show more basal sliding (due to an increased presence of meltwater) than those of the upper part, which might even show no basal sliding at all if the ice is frozen to the bedrock (due to lower temperatures in the highest regions). In addition, the upper area is notably shallower than the lower one (Fig. 3b), supporting the choice of a specific tuning parameter for this region. Two regions were therefore defined in order to distinguish the area where surface velocity observations are available (Fig. 3a) and the glacier is thick than the rest of the glacier (Fig. 3b). These two zones defining $c_l$ and $c_u$ roughly correspond to the averaged ablation and accumulation areas over the period 1947-2019; based on an equilibrium line altitude of $\approx$2900 m a.s.l..

### 3.1.3 Mass balance

Glacier Mass Balance (MB) was modeled on the basis of the difference between solid precipitation $P_s$ (accumulation) and melt $M$ (ablation), with correction factors $C_P$ and $C_M$:

$$\mathrm{MB} = C_P P_s - C_M M. \tag{2}$$

On one hand, solid precipitation $P_s$ was equal to precipitation $P$ when air temperature $T$ is below 0°C, and decreased to zero linearly between 0°C and 2°C. Here the air temperature $T$ and precipitation $P$ were taken from the closest MeteoSwiss weather station and adjusted using linear vertical lapse rates (Huss et al., 2009). On the other hand, melt $M$ was computed with a temperature index melt model (Hock, 1999; Huss et al., 2009):

$$M := \begin{cases} [f_\mathrm{M} + r_\mathrm{is}I]T & \text{if } T > 0, \\ 0 & \text{otherwise}, \end{cases} \tag{3}$$

where $f_\mathrm{M} = 1.787 \times 10^{-3}$, $r_\mathrm{i} = 2.146 \times 10^{-5}$ or $r_\mathrm{s} = 1.608 \times 10^{-5}$ ($r_\mathrm{is}$ is $r_\mathrm{i}$ or $r_\mathrm{s}$ if the surface is covered by ice or snow, respectively) are melt parameters taken from the modeling of Aletschgletscher over the period 1957-1980 (Jouvet et al., 2011), which is located near Gauligletscher, and $I(\mathbf{x}, j)$ is the direct solar radiation. In the absence of direct measurements of melt and accumulation, the uncertainty of $P_s$ and $M$ in (2) might be relatively high. To account for this, we introduced two correction factors $C_P$ and $C_M$.

### 3.1.4 Age computation and particle tracking

In addition to ice flow and mass balance equations, we also requested to compute the age of ice $a$ by solving the following transport equation (Gagliardini et al., 2013):

$$\frac{\partial a}{\partial t} + \mathbf{u} \cdot \nabla a = 1, \tag{4}$$

where $\mathbf{u}$ is the modeled velocity field, and $t$ is the time variable. The age of ice takes value 0 as the Dirichlet condition on the inflow boundary. For the sake of simplicity, $a$ was initialized everywhere to zero for the year 1947 as this was sufficient





to model the age of the ice formed after 1947. Alternatively, the use of a preliminary stationary model in 1947 could have permitted to extend the modeling of ages prior 1947.

Beside modeling the age of ice, the space-time position of the Dakota airplane and its pieces was inferred by integrating the modeled three-dimensional ice velocity fields by post-processing from their known space positions in 1947 and 2018, similar to Jouvet and Funk (2014) and Compagno et al. (2019).

### 3.1.5    Model calibration

An ensemble of model runs was performed with different parameters $A$, $c_l$, $c_u$, $C_P$ and $C_M$. The first three parameters controlled the ice flow strength, while the last two controlled the amount of precipitation and melt (or equivalently the mass balance

gradient). Additionally, we assumed that the correction melt factor $C_M$ takes two values $C_M = C_{M,<1980}$ and $C_M = C_{M,>1980}$ before and after 1980, respectively, as a DEM was available for year 1980. For the given $A$ and $C_P$, the calibration procedure consisted of several steps (summarized in Fig. 4) which were repeated until convergence was reached:

   i) The sliding parameter in the lower part $c_l$ was found to minimize the Root-Mean Square Error (RMSE) between modeled and observed velocities along 14 points located along the central flowline (Fig. 3a) during the period 2015-2019 after

170        modeling the 2010-2019 period, starting from the most recently available DEM (2010).

   ii) Parameter $C_{M,<1980}$ was found to minimize the RMSE between modeled and observed DEMs in 1980 after modeling the period 1947 to 1980.

   iii) Parameter $C_{M,>1980}$ was found to minimize the RMSE between modeled and observed DEMs in 2010 after modeling the period 1980 to 2010.

iv) The sliding parameter in the upper part $c_u$ is calibrated to get the best match between modeled and "observed" age of ice, where ice from 1960 was identified along line 3 (see Fig. 6 and Section 4.2). In the optimisation loop, $c_u$ is kept in the range [1,50] km MPa$^{-3}$ a$^{-1}$, which extends from nearly no sliding ($c_u$=1) to high basal motion ($c_u$=50).

Note that only 2 to 3 loops over the steps were necessary to achieve convergence of the parameter set. After calibration, we obtained 12 parameter sets (Fig. 4), which resulted from $A = 60, 100, 150$ MPa$^{-3}$ a$^{-1}$ and $C_P = 1, 1.2, 1.35, 1.5$. While the

calibration model runs used a 100 m resolution mesh to allow sequential multiple runs to be performed, we recomputed the 12 model runs with optimal parameter sets at a finer resolution (50 m) for further analysis.

### 3.2    Radionuclide sampling and analysis

### 3.2.1    Sample collection

In summer 2019, more than 200 surface samples were collected along five lines at the ablation zone of Gauligletscher (Fig.

5). Note that in summer 2018, 80 samples were collected as well, however, along a single central line (Fig. 5). Sampling lines were chosen to roughly follow flowlines whenever possible. Deviations from the planned sampling lines were allowed in the





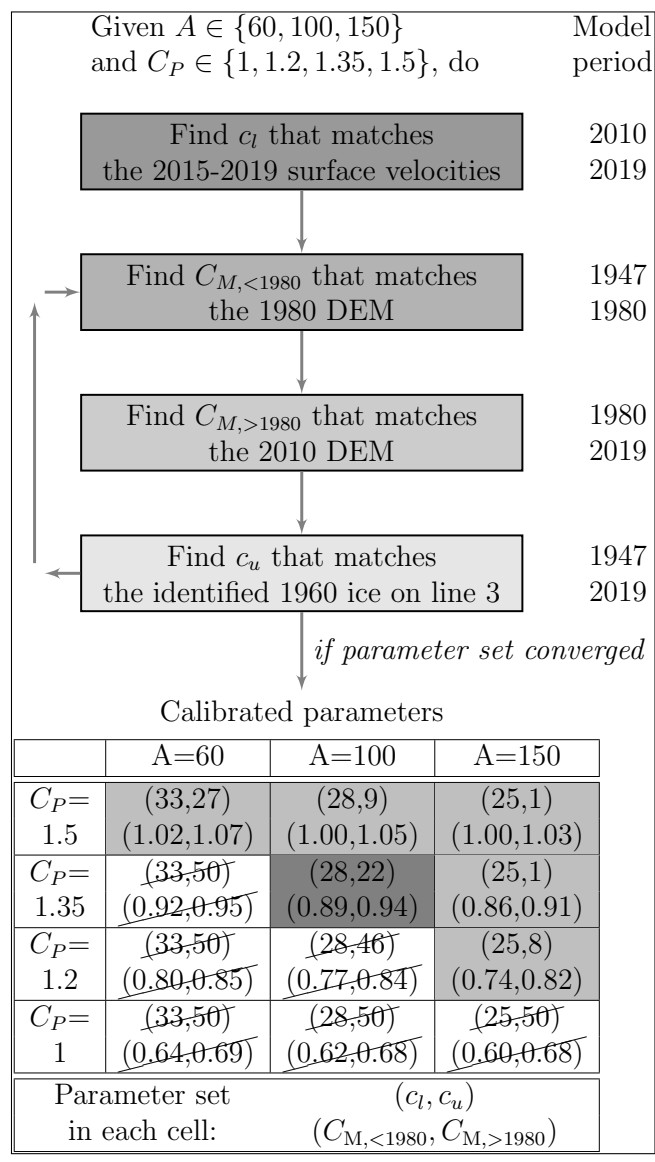

**Figure 4.** Calibration scheme (top) used in this study to find optimal parameters $c_l$, $c_u$, $C_{M,<1980}$, and $C_{M,>1980}$, for given $A$ and $C_P$ parameters, and parameter sets obtained after calibration (bottom). In practice, a very few iteration loops over the steps were sufficient to achieve convergence. The gray cells correspond to the parameter sets that were kept for analysis – the strike-through ones being discarded for yielding unrealistic results (Section 4.3). The dark gray cell indicates the closest to the mean of the 6 retained parameters, and was taken as the best-guess parameter set.

field in order to avoid crevasses. In summer 2019, the sampling area was determined to cover the 1954-1967 isochronal band derived from the original model by Compagno et al. (2019), with a high-resolution sampling zone (every ≈25 m) in the region



designated by the model, and a low-resolution sampling zone (every $\approx$50 m) further downstream and upstream for safety
reasons.

At each location, the sample ground was chosen to be located on visually dry ice at least 2 m away from surface meltwater
flow. The topmost layer ($\approx$30 cm) of ice was cleared, and then a minimum of 2 l of dry crushed ice was removed from the
glacier by means of a drill (8 cm, 1 m long). The samples were collected under sunny weather conditions, and mainly in
the morning with a minimum amount of meltwater flow on the glacier surface. The ice was left to melt for 2 days in the
polypropylene sample container which resulted in 1.2 l water. The GPS coordinates of all sampling locations were recorded.

### 3.2.2  Measurements

Uranium (U) and plutonium (Pu) were separated prior to any measurement activities. Details of the sample preparation is
described in Appendix B. The Mass Spectrometry (MS) analysis of Pu and U was carried out at Spiez Laboratory using a
multicollector inductively coupled plasma mass spectrometer (Neptune, Thermo Fisher). A self-aspirating PFA-ST nebulizer
and a SC-2DX autosampler (Elemental Scientific Incorporation) were used to introduce the samples. All MS measurements
were conducted in low resolution mode (R = 300).

For Pu measurements, a desolvating system (Aridus II, CETAC Technologies) was used to enhance the signal and to achieve
low hydride and oxide formation. In this configuration, the sensitivity for U was about $1 \times 10^8$ counts per second (cps) per
ng/ml $^{238}$U. All Pu isotopes were measured by SEM detectors. In order to correct for U and thorium (Th) interferences due to
hydride generation and tailing, 10 ng/ml U and Th standard solution were measured using Faraday detectors. Typical correc-
tion factors for the m/z 239, 240 were $3\times10^{-6}$, $5\times10^{-7}$ for $^{238}$U and $4\times10^{-8}$, $8\times10^{-9}$ for $^{232}$Th. Pu isotope concentrations
were calculated from the signal of the $^{242}$Pu tracer. The contributions of the Pu isotopes from the tracer were corrected math-
ematically based on the isotope ratios from the certificate. Considering the variations of these effects, the detection limit was
estimated to $2\times 10^{-3}$ mBq/kg $^{239}$Pu. The total acquisition time per sample was 3 min.
For U isotope ratio measurements, a Twinnabar (Glass Expansion) spray chamber was used. The sensitivity for U was about
$6 \times 10^6$ counts per second (cps) per ng/ml $^{238}$U. $^{235}$U and $^{238}$U were measured by means of Faraday detectors and the minor
isotopes by SEM detectors. For the measurement of $^{236}$U, a RPQ filter was used to lower the abundance sensitivity. The U
and Th tailing was corrected using a baseline subtraction. Hydride interference was corrected mathematically. Considering
these effects, the detection limit was estimated to $3\times 10^{-5}$ mBq/kg U$^{236}$. Standard sample bracketing using the IRMM-187
U isotope standard was applied for all measurements. IRMM-184 isotope standard was used for quality assurance. The total
acquisition time per sample was 18 min.



# 4 Results

## 4.1 Original model results

The original model from Compagno et al. (2019) was used to predict the most likely place on Gauligletscher where ice from
1954 to 1967 might appear – the period with the highest inputs of $^{239}$Pu in an ice core at Colle Gnifetti glacier, Switzerland
(Gabrieli et al., 2011). As a result, the model yields an asymmetrical U-shaped isochronal band located about 1 km above the
current glacier tongue (Fig. 5), which served to delineate the glacier region to be sampled (Section 3.2).

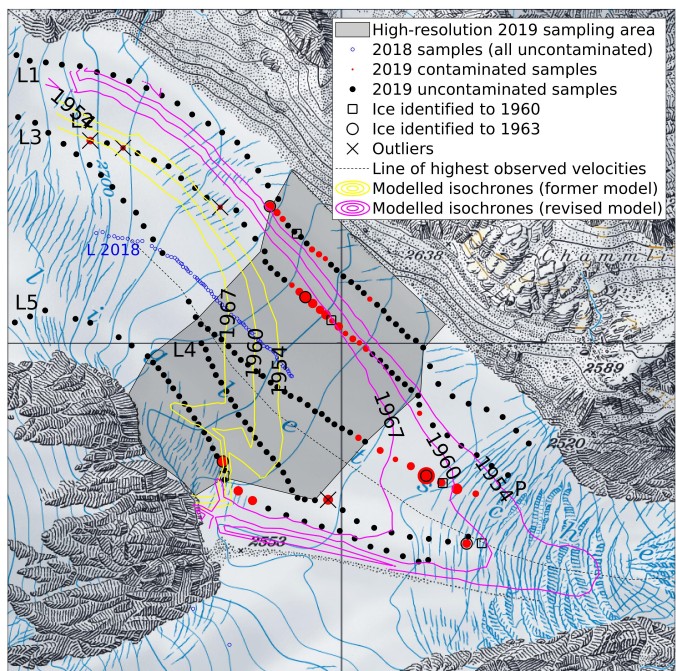

**Figure 5.** Topographic map of Gauligletscher showing the 1954, 1960, and 1967 isochrones inferred from the original (yellow contour lines)
and the revised best-guess (magenta contour lines) models, the five sampling lines (defined by 'L1' to 'L5', and indicated by black dots),
as well as the contaminated locations (where $^{239}$Pu activities exceed 0.25 mBq/kg) depicted by red dots, whose size corresponds with the
$^{239}$Pu activity. The squares and circles show the locations along each line identified as originating from 1960 and 1963 in Fig. 6, respectively.
Crosses indicate measurement outliers not included for analysis. The shaded area correspond to the region sampled in high resolution (i.e.,
≈25 m rather than ≈50 m). The dashed line shows the line of maximal observed velocities along each lateral transect. The blue circles mark
the line sampled in 2018, where no $^{239}$Pu activities > 0.1 mBq/kg have been found.





## 4.2   Dating with radionuclides

Figs. 5 and 6 show the $^{239}$Pu and $^{236}$U activities of the samples in plain view and along the five sampled lines, respectively.

The two tracers were found to be consistent in terms of pattern, and at the same time to indicate contaminated regions (above 0.25 mBq/kg in $^{239}$Pu, see Fig. 6). By contrast all the samples collected in summer 2018 show very small activities (below 0.1 mBq/kg in $^{239}$Pu, results not shown). Therefore, only 2019 sampling results were considered in the following.

Unfortunately, the contaminated region was captured only in the high-resolution sampling zone for lines 1 and 2, and is part of the low-resolution surrounding zone for other lines (Figs. 5 and 6). Following the footprint of $^{239}$Pu in the ice cores at Colle

Gnifetti (Gabrieli et al., 2011), Switzerland, and the eastern Tien Shan (Wang et al., 2017), Central Asia, the contaminated region was expected to correspond to the 1954-1967 period, which roughly extended from the first atmospheric NWT ($\approx$1953) to the last ones ($\approx$1963) when the USA and the USSR signed the Limited Test Ban Treaty (Fig. 1, Gabrieli et al., 2011). The highest $^{239}$Pu activity (found on line 2) shows $\approx$2.5 mBq/kg, which is about 2-3 times less than the highest ones found in the ice cores at Colle Gnifetti (7 mBq/kg), and the eastern Tien Shan (5.5 mBq/kg), however, it is slightly above the highest

activity (2.165 mBq/kg) measured in an ice core extracted from the Dome du Gouter, Mont Blanc, France (Tab. 7, Warneke et al., 2002).

Activities of $^{240}$Pu (not shown) follow a pattern very similar to the $^{239}$Pu ones, with a ratio $^{240}$Pu/$^{239}$Pu close to 0.2, which is a well-known value representing average fallout from US and USSR NWT (Krey et al., 1976). Note, however, that this only concerns samples with $^{239}$Pu activities above 0.25 mBq/kg as the $^{240}$Pu signal was disturbed by unknown interferences

for some samples with low $^{239}$Pu activities. We observe that $^{236}$U activities also follow a pattern similar to $^{239}$Pu (Fig. 6). Furthermore, the peak activity of $^{236}$U in line 3 ($\approx$ 1.6 $\mu$Bq/kg) is comparable with the one found in the ice core at the eastern Tien Shan glacier ($\approx$ 1.5 $\mu$Bq/kg)[1], which is given in Tab. 2 of (Wang et al., 2017). Nevertheless it must be stressed that interferences on the $^{236}$U signal caused the mass ratio of $^{236}$U/$^{239}$Pu to vary between 0.2 and 3. Therefore $^{236}$U was not further considered for dating purposes.

The contaminated band clearly shows a double peak structure for lines 1 and 5, a less pronounced one for line 3, multiple peaks for line 2, and a single dominant peak for line 4 (Fig. 6), after discarding the first one (see next paragraph). These two peaks are characteristic of the periods 1954-1958 and 1962-1963, respectively, matching the chronological order of NWT activities worldwide (Carter and Moghissi, 1977; Gabrieli et al., 2011; Wang et al., 2017). The first period spans more than $\approx$5 years and ends with a maximum in 1958, while the second period is shorter but more intense with a maximum in 1963.

We made use of this double peak structure to identify ice from 1963 – the highest and most recent peak – and ice from 1960 – the midway year between the two peaks characterized by a local minimum. The ice in 1963 is found at the highest and most upstream $^{239}$Pu activity location. It was easy to identify 1963 ice on all lines (Fig. 6) after filtering out a few outliers (see next paragraph). We associate this with a $\pm$2 year uncertainty, which corresponds to the width of the peak. On the other hand, 1960 ice was easily found between the two detected peaks of lines 1, 3, and 5. We associate it also with a $\pm$2 year uncertainty, which

corresponds to the distance between the two peaks. By contrast, the interpretation was more difficult for lines 2 and 4. Line 2

[1]Note that the $^{236}$U activity given in the last column of Tab. 2 of (Wang et al., 2017) has an erroneous unit: it should read $\mu$Bq/kg instead of mBq/kg.





shows 3 peaks above 0.25 mBq/kg, which might be caused by the near alignment of the sampling line with the isochrone (Fig. 5). We chose midway between the two highest peaks in $^{239}$Pu, but increased the uncertainty to $\pm 6.5$ years, which corresponds to the length of the 1954-1967 period. By contrast, the high end of line 4 coincides with the highest observed velocities (Fig. 3) indicating that the 1960 position sought along this line is expected to be the most downstream one of all five lines. This

statement is supported by subsequent modeling results (Fig. 5). By comparing the $^{239}$Pu activities found on lines 3 and 4, we find a similar pattern for the presumed 1962-1963 peak (Fig. 6), the other peak associated with 1954-1958 likely being missed due to its location further downstream of the sampled area. We therefore inferred the 1960 position along line 4 by analogy with the one found on line 3 (Fig. 6).

Let us note that four samples with $^{239}$Pu activities above $\approx$0.25 mBq/kg were found away from any identified contaminated

areas (see 'outliers' in Figs. 5 and 6). Two of these outliers (on line 2) have activities just above the threshold value. This justifies their disqualification. However, there remain two isolated outliers on lines 2 and 4 with activities of $\approx$0.5 mBq/kg and $\approx$1 mBq/kg, respectively, which might be caused by sampling or measurement errors. Note that the unexpected isolated high activity seen on line 4 might be due to the alignment of the sampling line with the isochrone.

### 4.3  Revised model results

Fig. 5 shows that the 1960 and 1963 ice revealed by radionuclide contamination is quite far downstream of the ice predicted by the original model (Compagno et al., 2019). To investigate this, we explored further ice flow ($A$, $c_l$, $c_u$) and mass balance ($C_P$ and $C_M$) parameters on the basis of 12 new model runs (Fig. 4, bottom panel).

Parameters $A$ and $c_l$ were found to match observed surface velocities of the ablation area over the 2015-2019 period (Figs.

3a and 7c). As a result, the inferred parameters led to higher $c_l$ (i.e., higher sliding velocities) as compared to parameter $c_l = c_u = 12.5$ km MPa$^{-3}$ a$^{-1}$ of the original model by Compagno et al. (2019) with $A = 60$ MPa$^{-3}$ a$^{-1}$. This is due to the new dataset of observed velocities (Appendix A), which showed faster ice in the ablation area than the dataset used formerly. On the other hand, $C_{M,<1980}$ and $C_{M,>1980}$ were found to match observed DEMs in 1980 and 2010 for all parameters $A \in \{60, 100, 150\}$ and $C_P \in \{1, 1.20, 1.35, 1.5\}$ (Fig. 7). This is a major difference with our former model (Compagno et al.,

2019), which used only $C_P = 1$. Thus we explored a stronger mass balance vertical gradient, i.e., higher precipitation and higher melt scenarios, as no direct measurements were available there. Lastly, in the absence of direct ice flow measurements in the accumulation area, we adjusted the sliding parameter there ($c_u$) to match the age of ice identified by radionuclide contamination.

As a result of this calibration, 6 of the 12 model runs (see shaded cells in Fig. 4) provided results which were far more

realistic mainly with $C_P = 1.35$ and $C_P = 1.5$, i.e., with a 35-50% additional precipitation correction than the ones obtained using $C_P = 1$:





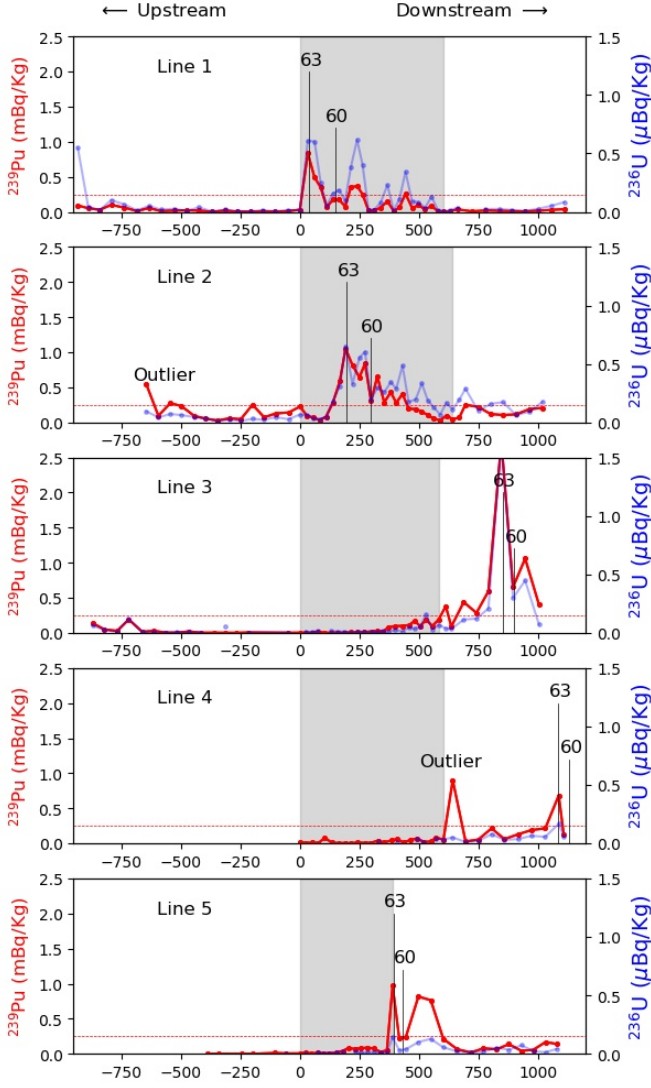

**Figure 6.** Activities of $^{239}$Pu and $^{236}$U along the five sampled lines as defined in Fig. 5 from upstream to downstream. The shaded area correspond to the region sampled in high resolution (i.e., $\approx$25 m instead of $\approx$50 m). The vertical line depicts the position where ice from 1960 and 1963 have been identified ($\pm$2 or $\pm$ 6.5 years). The threshold 0.25 mBq/kg used to qualify contaminated samples in Fig. 5 is indicated by a dashed horizontal line. Note that the uncertainty associated to $^{239}$Pu measurement is less than 4% for all samples that are above 0.1 mBq/kg.

i) All model runs except one based on the two lowest $C_P$ values led to higher sliding in the upper zone ($c_u$) than in the lower zone ($c_l$), which was an unrealistic situation, as sliding was expected to increase at lower elevations (due to the presence of further meltwater and temperate ice).





ii) The value of $C_P = 1.35, 1.5$ always led to a substantially improved agreement between observed and modeled glacier
length evolution (Fig. 7d).

iii) To a lower extent, $C_P = 1.35, 1.5$ led to a better match between observations and modeling in terms of DEMs and
2015-2019 surface velocities (Figs. 7a, 7b, and 7d).

Based on these findings, we discarded 6 model runs that yielded unrealistic situations (strike-through cells in Fig. 4), and kept
only the 6 remaining ones for further analysis (shaded cells in Fig. 4).

Interestingly, the parameter set that was the closest to the mean of the six retained sets showed values: $A = 100$ MPa$^{-3}$
a$^{-1}$, $(c_l, c_u) = (28, 22)$ km MPa$^{-3}$ a$^{-1}$, $C_P = 1.35$, $C_{M,<1980} = 0.89$ and $C_{M,>1980} = 0.94$ (Fig. 4), which were very similar
to the values calibrated for Aletschgletscher in another study (Jouvet et al., 2011) – namely $A = 100$ MPa$^{-3}$ a$^{-1}$, $(c_l, c_u) =$
$(23.3, 0)$ km MPa$^{-3}$ a$^{-1}$, $C_P = 1.35$ and $C_M = 1$. Note that unlike Gauligletscher, the mass balance parameters calibrated for
Aletschgletscher were based on direct measurements of melt and accumulation. We therefore selected this parameter set as
being our best-guess one.

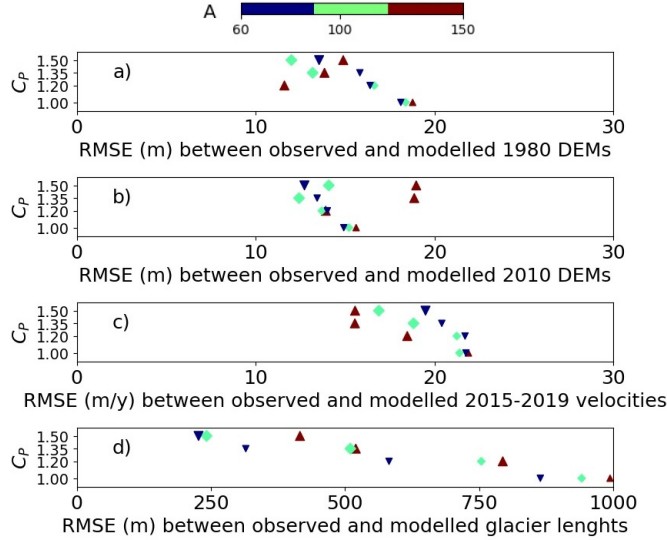

**Figure 7.** Root Mean Square Error (RMSE) between modeled and observed variables for the 12 model runs (indexed by $C_P$ and $A$) relating
to the 1980 DEM (a), the 2010 DEM (b), the 2015-2019 surface velocities(c), and the length of the glacier tongue (d). Large (or small)
symbols indicate the parameter sets that yield realistic (or unrealistic) results, and then were kept (or discarded) for analysis (Fig. 4).

Fig. 8 shows the modeled age of ice at the location where ice was identified as originating from 1960 and 1963. As a result,
we get a much better agreement between modeled and observed 1960 and 1963 age at lines 3 and 4 when comparing new to
former model results. Note that this was expected for line 4 as the model was adjusted to give the best agreement for 1960.
Focusing on line 3 that has the narrowest interval of confidence, the 6 model runs revealed a very good fit. Fig. 5 displayed the

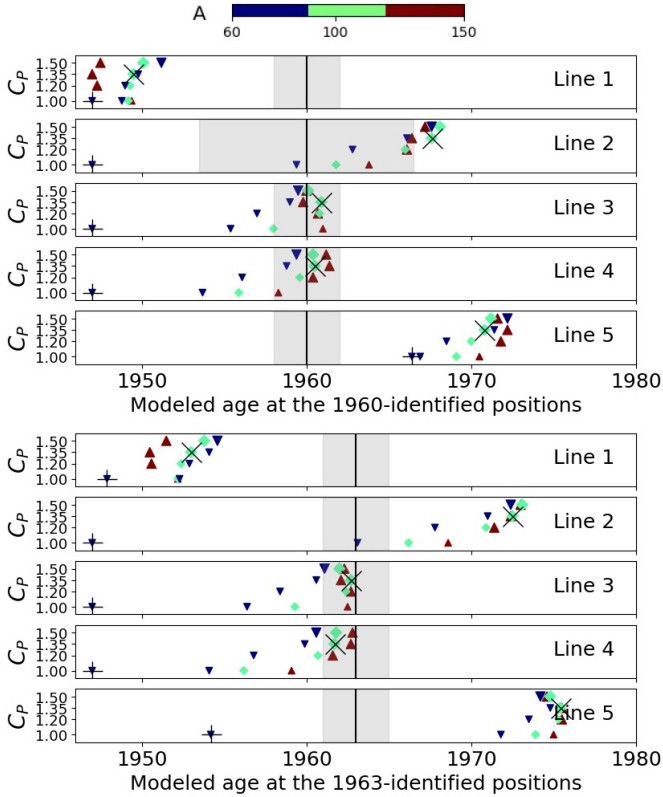

**Figure 8.** Modeled age of ice at the location where ice has been identified as originating from 1960 (top panels) and 1963 (bottom panels) in Fig. 6 for the 12 new model runs. The gray area represents the interval of confidence. Symbols + and × designate the results of the original model by Compagno et al. (2019), and the best-guess model run among the 12 (Fig. 4), respectively. Large (or small) symbols indicate the parameter sets that yielded realistic (or unrealistic) results, and then were kept (or discarded) for analysis.

1960 isochrone in plain view obtained using the original model and the revised best-guess one, and confirms the substantial improvement achieved for matching radionuclide-based dated ice. In contrast, a relatively high mismatch between observation and modeling was found for side lines 1 and 5. Indeed, modeled age at the 1960 and 1963 identified positions on line 1 was largely overestimated (more than a decade), while the one for line 5 was underestimated by the same amount (Fig. 8). Fig. 5

illustrates the model-observation mismatch in terms of isochrones: The model tends to exaggerate the asymmetry in the ice flow (visible on Fig. 3) resulting in too-slow ice (and then too old ice) in the north-east side, and too fast ice (and then too young) on the south-west side. As the flow asymmetry was directly controlled by the bedrock topography (Fig. 3b), it is likely that this mismatch can be corrected by adjusting the bedrock lateral profile, i.e., making it deeper where ice is found to be too old/slow, and vice-versa. To a leaser extent, the discrepancy between modeled and observed age of ice on line 2 can be

explained in a similar way.



Thanks to the newly calibrated model, we could derive the age of ice everywhere at the surface of Gauligletscher (Fig. 9). The results show that most of the current ablation zone released ice prior to 1950. Note that our map is incomplete as we did not model ice older than 1947 – the year of initialization. Interestingly, the most recent modeled isochrones (after 1970) are U-shaped while the much older ones (before 1970) are V-shaped. The peaks of the U and V roughly coincide with the maximal

velocity across glacier lateral transects (Fig. 9). The asymmetry between the left and right lateral margins was quite strong, and probably stronger than it is in reality due to an inaccurate bedrock as discussed before.

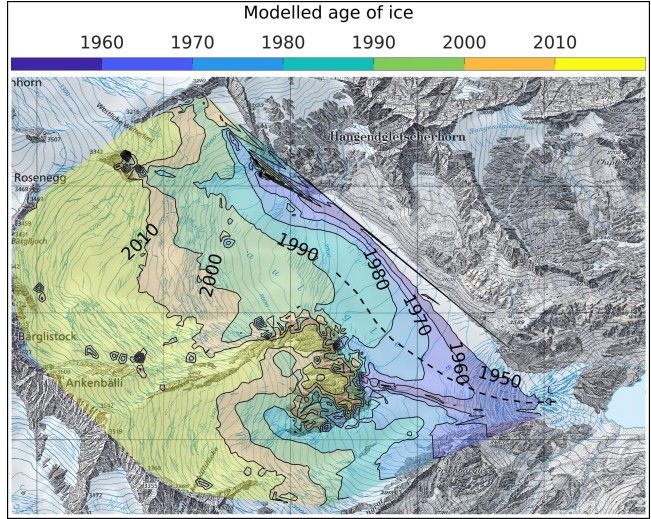

**Figure 9.** Topographic map of Gauligletscher showing the modeled age of ice at the surface using the best-guess model run (Fig. 4). The dashed line shows the line of maximal observed velocities along each lateral transect.

## 5   Discussion

The results of the sampling campaign demonstrate that the region featuring higher $^{239}$Pu, $^{240}$Pu and $^{236}$U activities coincides with the area where ice originates from the early 1950s to the late 1960s:

– Contamination in both radionuclides has the same pattern and for the most part appears band-wise (Fig. 5). The sampled lines show ≈300 m-long bands with $^{239}$Pu activities mostly above 0.25 mBq/kg (Fig. 6) up to 2.5 mBq/kg, which is in line with the highest activities found in three different ice cores (≈2.2, 5.5 and 7 mBq/kg). These contaminated bands presumably reflect the most intense period in terms of NWT, namely 1954-1967 (Gabrieli et al., 2011). As this period lasted roughly 13 years, this corresponds to an ice flow of ≈23 m/y, which is in the high end of the 2015-2018 ice flow

observations along those lines (13-22 m/y, see Fig. 3), however, the glacier was most likely faster in the past (as thicker). By contrast, off-band samples show very low activities (under 0.1 mBq/kg) with few exceptions (Fig. 6). Note that the





exact peak activity year (1958 or 1963) might have been missed as the sampling distance (≈25-50 m) corresponds to ≈1-3 y time interval.

– The contaminated part of line 2 is located slightly above the location where pieces from the Dakota airplane were found in 2018 (Fig. 3a). Regardless of the location where the pieces were absorbed by the glacier, the ones released in 2018 are located on an isochrone associated with year 1947, i.e., 5 years before the period where contamination should be observed. The position of the contaminated band and of the released pieces is therefore consistent, since the surface ice is expected to be older downstream than upstream (Fig. 1).

– Three lines (1, 2 and 5) show a double peak structure in $^{239}$Pu, $^{240}$Pu and $^{236}$U activities, which are typical of the years 1957 and 1963 (Wang et al., 2017). The spacing between the two peaks (100-150 m, Fig. 6) is again consistent with an ice flow speed of ≈20 m/y and the 5-7 year interval. The double peaks of lines 3 and 4 were missed most likely because of insufficient sampling resolution and too-small sampling area (the presence of large crevasses prevented us from sampling further downstream).

– The shape of the contaminated bands is consistent with the observed ice flow (Fig. 3a): the contaminated bands of central flowlines are shifted downstream compared to the side ones, and the lowermost one is located on the profile where observed velocities are maximal.

– Ratio $^{240}$Pu/$^{239}$Pu was found for the most part close to 0.2, which is a well-known value representing average fallout from the US and USSR NWT (Krey et al., 1976).

– Another line located upstream of the contaminated region was sampled and analyzed in summer 2018 (results not shown). No $^{239}$Pu activity above 0.1 mBq/kg was detected, similar to the upper part of line 3 in 2019, which is located nearby. Corroborating the 2018 and 2019 sampling supports the fact that the radionuclide contamination is stable in space and time.

Based on this evidence, we identified ice from 1960 and 1963 with an accuracy of a few years. To our knowledge, this is the first time that plutonium-based tracers have been used to date ice in the ablation zone of an Alpine glacier.

Our original model (Compagno et al., 2019) predicted ice from 1954-1967 located mostly upstream of the ice revealed by $^{239}$Pu, $^{240}$Pu and $^{236}$U contamination (Fig. 5). To correct the model, we enlarged the parameter space to test further ice flow and mass balance regimes, and recalibrated our model accordingly (Fig. 4). The newly calibrated model permits a significantly better agreement to be achieved between the modeled and the radionuclide-based 1960 and 1963 ice (Fig. 8). The new data can even serve to further constrain our parameter space: the 6 model runs based on low precipitation rates delivered unrealistic situations, and then were discarded. As a result of the recalibration, we found that ice flow, precipitation, and melt were underestimated in our original model. Interestingly, the best-guess parameter set is very similar to the optimal parameter set found for modeling Aletschgletscher from 1880 to 1999 (Jouvet et al., 2011), all of these parameters being calibrated against measurements and validated by means of an independent tracer (Jouvet and Funk, 2014).





The sparsity of data used for tuning the ice flow model is a well-known source of model uncertainty. Indeed, observed surface velocities used to adjust key parameters only cover restricted areas and time frames (i.e., mostly the ablation area for the recent past) due to limitations in remote sensing methods. Indeed, snow-covered accumulation areas show hardly any features that can be tracked, and there is hardly any usable satellite data prior to the 1990s. Thus our model might poorly reproduce the ice flow in the distant past and/or over the accumulation area. Furthermore, several ice flow and mass balance regimes (from slow ice and low mass balance gradient to fast ice and high mass balance gradient) can reproduce observed DEMs in a similar fashion

– an ambiguity that sparse data can not resolve. By contrast, tracers such as radionuclides yield much more global data (in space and time) as they result from the long-term glacier movement and mass balance. Tracers further reflect the ice dynamics over the entire glacier thickness – the surface isochrones being different if ice particles have deep or shallow trajectories. From a modeling perspective, such datable tracers (or more generally Lagrangian data) are therefore extremely valuable, as they potentially can constrain the space of model parameters much better than a modern data set of observed ice velocities can do.

Here, radionuclide-based data also permitted precipitation and melt upwards in the mass balance models to be substantially revaluated.

    Our results also shed light on the influence of the bedrock topography on isochrones. Our bedrock was inferred by interpolating ice thickness profiles obtained by Ground Penetrating Radar (GPR) (Rutishauser et al., 2016). Therefore, the bedrock data might be locally inaccurate if the reflectance is not well interpreted or due to interpolation errors. Here we have shown that

our bedrock is likely too shallow on the north-east side and too deep on the other side. For this reason, surface isochrones such as the ones revealed here by plutonium contamination can be used to optimize the bedrock, in particular by correcting its shape along lateral glacier sections. While surface observed ice flow velocity or mass balance are data commonly used to interpolate GPR-based measured profiles, and to derive entire bedrock elevation distribution (e.g., using inverse modeling, Morlighem et al., 2011), the use of Lagrangian data could provide new insights and should be investigated further.

Here we investigated anthropogenic $^{239}$Pu, $^{240}$Pu and $^{236}$U contamination in order to discover isochrones of relevance for the release of ice in the ablation zone of a temperate Alpine glacier, however, other tracers might be worth considering as well. Using atmospheric NWT related to the 1954-1967 period is suitable for dynamic Alpine glaciers for several reasons. First, typical time scale for the transportation of ice particles from the top to the bottom varies from a few decades to a few centuries (according to the glacier size and dynamics) and the most relevant period (1954-1967) is ≈60 years old. Second,

this period extends over more than a decade with a typical double-peak structure, which makes it easily detectable. Last, in general climate conditions during the 1954-1967 period were characterized by significant net accumulation layers (in contrast with Post-1980s) favouring the preservation of the signal left by the fallouts of atmospheric NWT. By contrast, sporadic events such as the nuclear accident at Chernobyl (Haeberli et al., 1988), Saharan dustfalls (Wagenbach and Geis, 1989) and volcanic events (Kellerhals et al., 2010) which have been detected in ice cores located in accumulation zones, might be too short-lived

to be detected at the surface of the ablation area. The main challenge for finding alternative tracers is to locate some that have not been washed away by percolating meltwater in the ablation area. Here we found that the $^{239}$Pu signal at the surface of Gauligletscher is very close to the activity found within an ice core extracted near the Mont Blanc (Warneke et al., 2002), and about 3 times less to the activity found within an ice core extracted at Colle Gnifetti (Gabrieli et al., 2011).





## 6 Implications for the Dakota

Using an early version of the model (Compagno et al., 2019), we estimated both the current position and the future emergence of the Dakota airplane, which crash-landed on Gauligletscher in 1946 and was subsequently buried within the ice (Section 2). Our new model calibrated with radioisotope data permits a significant update of this estimate. According to our best-guess model run (Fig. 4), the Dakota would have reached a maximum depth of 120 m under the surface in the late 1980s and would have emerged at the surface around 2015 slightly downstream from the place where pieces were found between 2012 and

2018 (Fig. 10, Panel a). This indicates that the main body of the Dakota most likely lies in this region close to the ice surface, and will emerge there in the coming years. This new model result is in stark contrast to our previous one, which estimated a trajectory of roughly half this length (Fig. 10, Panel a) and half this depth (the maximal depth computed by Compagno et al. (2019) was 68 m). This illustrates the significant underestimation of ice fluxes in the former model, and demonstrates the great added-value of the newly-used Lagrangian data to refine our modelling of the mass balance and the ice flow.

Our new estimate still upholds the fact that certain pieces of the aircraft (e.g. one engine) became dislodged from the fuselage in 1947, which explains why each component followed a different trajectory. However, it refutes the theory that the wreckage, which had already been found, were moved far away from the main fuselage in 1947 (Compagno et al., 2019). Indeed, integrating the modeled velocity field backward-in-time starting from the location where wreckage was found in 2018 yields a trajectory that is slightly shorter but comparable to the one obtained by forward-in-time integration (Fig. 10, compare

Panels a and b). By contrast, the backward and forward trajectories computed by Compagno et al. (2019) were significantly different (Fig. 10, compare dashed lines in Panels a and b). If the model were free of any numerical or physical errors, and if the entire Dakota aircraft had been released from the ice in 2018, then the two trajectories (forward and backward) would coincide. Therefore, the trajectory discrepancy can be understood as a measure of combined model errors and the distance between the body of the Dakota and the pieces dislodged from it in 1947. The relative minor discrepancy between trajectories from the new

model shows that the Dakota main body and the pieces most likely always remained in close proximity to one another.

Among the five other selected model runs (Fig. 4), two show a very similar trajectory (with 2019 positions lying with a maximal distance of 100 m, see Figs. 10 and 11). Conversely, the three other model runs that use parameter $A = 150\,\mathrm{MPa}^{-3}\,\mathrm{a}^{-1}$ show shorter trajectories with end positions 300 to 500 m from the position of the released pieces (Fig. 11). This discrepancy (up to 20% of the trajectory) is caused by slower velocities of the airplane in the early stage in the highest area. Indeed, sliding

is nearly suppressed in the accumulation zone (i.e., $c_u$ is close to zero, Fig. 4) as a result of the calibration procedure in the case of the softest ice $A = 150\,\mathrm{MPa}^{-3}\,\mathrm{a}^{-1}$ in order to maintain the ice flux. In other words, the higher vertical deformation rate of ice is compensated by the lower basal motion. Therefore, the ice velocities are low in the early stage of the movement of the submerged aircraft such that it takes longer for it to travel to a fast-moving area, a fact which causes discrepancies in the calculated trajectory length. This reveals the sensitivity of our trajectory estimate to uncertain parameters, and shows that an

error margin of several hundreds of meters should be taken into account (Fig. 11).

Beside the modeling results, we also computed the backward trajectory by integrating the observed horizontal velocity field (Fig. 3, Panel a), assuming the flow field in the current era times (here 2015-2019) to be representative of the 1947-2018 period.

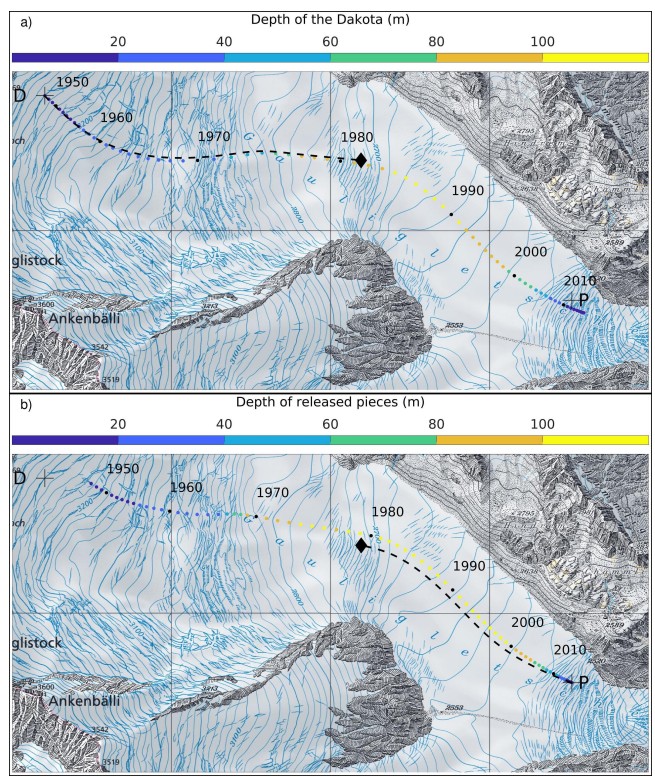

**Figure 10.** Topographic map of Gauligletscher showing the modeled horizontal trajectory of a) the main body of the Dakota from 1947 to 2019 after integrating the modeled velocity field forward-in-time; and b) the airplane pieces released from 2018 to 1947 after integrating the modeled velocity field backward-in-time. In both cases, we used the best-guess model run (Fig. 4). The corresponding forward (or backward) trajectory and its end position in 2019 (or 1947) computed by Compagno et al. (2019) is shown with a dashed line and a diamond symbol in Panel a (or b) for comparison purposes. The location where the Dakota airplane crash-landed in 1946, and where pieces of the plane emerged in 2018 at the surface are indicated by the letters 'D' and 'P', respectively.

As a result, we found a backward trajectory (not shown) very similar to the one obtained by Compagno et al. (2019) as depicted in Fig. 10, Panel b. This trajectory is therefore unrealistically too short (by a factor of approx. 2). This simple estimate – whose advantage is to be free of any modeling considerations – fails because ice flow was significantly faster in the past; however, no direct observation is available to take this into account. This illustrates the incontrovertible superiority of an ice flow model that integrates information on the long-term ice motion (such as Lagrangian data used here) to reliably model the space-time trajectory followed by ice particles.



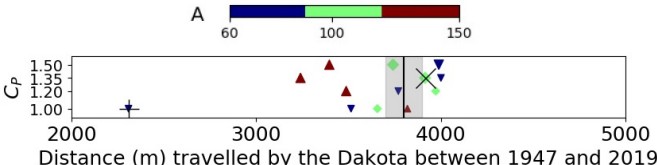

**Figure 11.** Distance traveled by the Dakota main body along the trajectory line drawn on Fig. 10 (Panel a) for the 12 model runs. The vertical line shows the average position of the pieces that were found in 2012, 2015 and 2018, and the shaded area shows its confidence taking into account the delay between the findings and the spread of the objects. Symbols + and × designate the results from the original model by Compagno et al. (2019), and the best-guess model run among the 12 (Fig. 4), respectively. Large (or small) symbols indicate the parameter sets that yielded realistic (or unrealistic) results, and then were kept (or discarded) for analysis.

## 7 Conclusions

In this study, we successfully used plutonium contamination induced by atmospheric nuclear weapon tests performed in the 1950s and 1960s to date the ice of the ablation zone of an Alpine glacier. While this tracer has been used multiple times for dating ice cores and other archives, this is the first time it has been used to date ice transported within the gravitational flow of ice to the ablation area, and that might have been deteriorated by temperate conditions and/or percolation water. Here we showed that the signal is still well preserved and that well-known activity peaks (i.e., 1963) can be identified, giving rise to positions that are in line with the observed ice flow of Gauligletscher. The combining of radionuclide contamination, state-of-the-art ice flow and mass balance modeling allowed us to derive a complete age map with an unprecedented degree of accuracy for a mountain glacier – a technique that is applicable to other glaciers. Our study sheds light on the potential of our method in two different ways:

- Radionuclide-based tracers provide invaluable data for model calibration as they integrate information on the long-term ice dynamics and mass balance over the entire glacier. Such Lagrangian data outperform the sparse surface ice velocities and mass balance data that are usually only available and used for that purpose. They can therefore much better constrain the model inputs (including the bedrock) as demonstrated here. It is remarkable to note that the optimal parameter set we found here is very close to the one optimized for Aletschgletscher, which is a much bigger glacier. Therefore, we recommend using this parameter set for modeling further temperate medium to large-size glaciers of the Alps.

- The accurate and complete mapping of the age of ice is highly relevant to perform "horizontal ice cores", i.e., for sampling surface ice of the past chronologically without having for performing expensive and logistically complex deep ice core drilling. Note, however, that only a limited number of proxies traditionally used in high-alpine ice cores are expected to remain identifiable at the glacier surface because of alterations due to ice thermo-mechanical changes. Here isochrones were found to be nearly aligned with the flow near the north-west side, and very tight towards the sides, which called for the revision of our sampling strategy. Indeed, sampling in the lateral direction (i.e., orthogonal to the



isochrones) would facilitate the identification of the double peak structure in this region, and subsequently facilitate the derivation of an age-location relationship. Last, our dating method could also serve to identify the origin of any relics that have been transported by ice flow over decades and been released by the ice, and found at the surface.

Furthermore, by including the radioisotope-based dating, it becomes possible to improve our estimate of the emergence time
and position of the Dakota airplane, which crash-landed on Gauligletscher in 1946, and is still trapped within the ice. Indeed, the revised model shows that the Dakota is most likely going to emerge rather soon downstream from the place where pieces have been found recently.

*Data availability.* Radionuclide data will be published upon acceptance of the paper.

## Appendix A:  Surface observed velocities

To estimate of the surface velocity of Gauligletscher, we used the Matlab toolbox ImGRAFT (Messerli and Grinsted, 2015) to derive ice displacement fields by template matching (we used here normalized cross-correlation techniques) of satellite yearly-spaced pairs of ortho-images between 2015 and 2019. Artifacts were filtered based on signal-to-noise ratio, velocity magnitude or direction, and consistency between data sets (obtained from pairs 2015-2016, 2016-2017, 2017-2018, 2018-2019) assuming constant dynamic during the period 2015-2019. Note that we only used orthoimages of the end of the melt
season to minimize the snow-covered areas where features can hardly be detected. The final velocity field (Fig. 3) was obtained by taking the median value between the data sets when the standard deviation was found sufficiently small. The velocity data covers the ablation area only as the accumulation area does not show enough features to apply the tracking algorithm. Note that the resulting observed velocities were found consistent with another independent data set (Dehecq et al., 2019) based on repeat-image feature tracking (Dehecq et al., 2015).

## Appendix B:  Sample preparation

### Reagents and equipments

All reagents were of analytical grade. Ultrapure water (18.2 MΩcm, Barnstead), 65 % nitric acid p.a. (prepared on a quartz sub-boiling apparatus), 40 % hydrofluoric acid suprapur, 37 % hydrochloric acid suprapur and ammonium iron (II) sulphate hexahydrate pro analysis (all from Merck) were used for the analyses. Standard solutions of U, Th, Indium (all from Alfa
Aesar, Germany) were used to determine the correction factors due to tailing and hydroxide formation. U isotope standard IRMM-187 and IRMM-184 (all from Joint Research Centre, Belgium) were used for isotope ratio calibration and validation. A standard solutions of $^{242}$Pu was used as radiochemical yield tracer. The extraction chromatography resins TEVA and UTEVA were obtained from Triskem International (Bruz, France). All the resins have $100 - 150 \ \mu$m particle sizes. All separations were made with a vacuum box. All measurements were performed on a multicollector inductively coupled plasma mass spectrometer



(MC-ICP-MS, Neptune, Thermo Fisher).

**Separation of iron hydroxide precipitation**

To 1 kg of water, 300 $\mu$l of $^{242}$Pu radiochemical yield tracer (conc. $1.10^{-12}$ g/g) and 240 mg of $FeCl_3$ (hexahydrate) was
added. The mixture was warmed up and stirred at 60 °C for 30 min. Ammonia (25% aq. solution) was added until pH 8 was

reached. After 30 min, the mixture was cooled down and allowed to precipitate. The supernatant was extracted by suction, the
precipitate was transferred, centrifuged and the supernatant was again extracted by suction. The precipitate was dissolved with
$HNO_3$ (30 mL, 4.5 M) before $(NH_4)_2\,Fe(SO_4)_2 \cdot H_2O$ (500 mg) was added. The solution was vigorously shaken and allowed
to stand for 10 min. Pu and U were separated sequentially on a TEVA and UTEVA extraction chromatography resin.

**Separation of plutonium**

The TEVA column was conditioned with 10 mL of a 0.2% $HNO_3$/0.002% HF/0.01 mM Fe(II) solution and then with 10 mL of
3 M $HNO_3$/0.1 mM Fe(II) solution. After loading the sample, the TEVA column was rinsed with 25 mL of 6 M HCl (remove
Th) and then with 50 mL of 3 M $HNO_3$/0.1 mM Fe(II) solution. The Pu fraction was eluted with 20 mL of a 0.2% $HNO_3$/0.2%
HF/0.01 mM Fe(II) solution and measured with MC-ICP-MS.


**Separation of uranium**

The UTEVA column was conditioned with 10 mL of a 0.2% $HNO_3$ solution and then with 5 mL of 3 M $HNO_3$ solution. After
loading the breakthrough from the TEVA column, the UTEVA column was rinsed with 10 mL 6 M HCl and 10 mL 3 M $HNO_3$.
The U fraction was eluted with 20 mL of 0.2 % $HNO_3$/0.002 % HF solution.


*Author contributions.* GJ designed the study, ran the simulations, and wrote most of the manuscript except Section 3.2.2 and Appendix B,
which were written by SR, and Section 3.2.1, which was written by LG. SR, HS, and JC supervised the analytical team and performed
analyses at the Spiez Laboratory, the Swiss Federal Institute for NBC-Protection. LG organized military logistics and fieldwork, supervised
the collection of samples in summer 2019 and supported the laboratory analytics team. DS initiated collaboration between academic partners,

Spiez Laboratory, NBC Defense Laboratory 1 and The Swiss Armed Forces Alpine Command. DS organized the fieldwork and the sample
collection in summer 2018. LC performed the original model runs that served to locate ice from the 1960s prior to the sampling campaigns.
All authors contributed to the study.

*Competing interests.* The authors declare that they have no conflict of interest.



*Acknowledgements.* We are thankful to Lt Col Roger Herger and all the members of the radiochemistry group from the 2nd Company of the Swiss Armed Forces NBC Defence Laboratory 1 for extensive sample preparation, radiochemical processing, and sample analysis. We thank Major Patrick Bargsten, who launched the idea to seek for traces related to NWT in surface ice samples. The Swiss Armed Forces Alpine Command is acknowledged for its generous support during the fieldwork. The authors thank the Swiss Air Force for airlifts and material transport flights to Gauligletscher, Heinz Gäggeler and Theo Jenk for useful comments on the manuscript, Olivier Gagliardini and

Joël Morgenthaler for helping with the Elmer/Ice age of ice solver, Eef van Dongen for support with Elmer/Ice, Matthias Huss for advice on the mass balance modeling, Julien Seguinot for processing Sentinel-2A satellite images with SentinelFlow that were used for feature-tracking in an early version, Amaury Dehecq for giving access to a second data set of observed ice flow velocities used here for comparison purposes, and to Susan Braun-Clarke for proofreading the English. The sampling area was defined during fieldwork preparation while the first author was working at the Laboratory of Hydraulics, Hydrology and Glaciology (ETHZ).



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
