# Peer review of "Mapping the age of ice of Gauligletscher combining surface radionuclide contamination and ice flow modeling"

_The Cryosphere, 2020_

## Referee Comment (RC1) · Anonymous Referee #1 · 8 Jul 2020

*Review of « Dating the ice of Gauligletscher, Switzerland, based on surface radionuclide contamination and ice flow modeling » submitted to The Cryosphere by Jouvet et al.*

In this study the authors use measurements of radionuclide activity in ice samples taken from the surface of  Gauligletscher to calibrate an ice flow model. Radionuclide activity anomalies are typical of the nuclear weapon tests conducted in the 1950s and 1960s and provide a constraint on the ice age.
The model is then used to update the trajectory of an aircraft which crash-landed on the glacier in 1946 and parts of which have recently reappeared. The predicted trajectory appears to be more consistent than that given by a previous study.

In addition to the radionuclide data, 2 DEMs and a velocity field are used to constrain the model parameters (3 parameters for ice dynamics (A,cu,cl) and 2 parameters for surface mass balance (Cp and CM)).

This is a very original study, the results are convincing and the paper is well presented.

I have listed below some general and detailed comments that deserve some considerations by the authors :

***General comments :***
- The parameterization used in this paper for the basal friction differs from the previous study since here 2 distinct coefficients are used, for the upper and lower parts of the glacier. The friction coefficient for the upper glacier is calibrated using the radionuclide measurements and appears to be consistent with what we could geuss as it leads to low sliding in the upper part of the glacier. Thus I wonder what is the real contribution of the radionuclide data for the calibration of the model, i.e. is it not possible to calibrate this parameter without these data and/or what would the model give if we assume, for example, no sliding above the equilibrium line?

- The performances of the model are presented only in terms of RMSE, as the parameterizations used are quite simple I think it could be interesting to show some error maps (speed and elevation) in order to discuss the robustness of the model. Notably only the speeds and ages along the centre line are used for the calibration, which corresponds roughly to the aircraft's trajectory. However the ages on the sides of the glacier are less well reproduced by the model and it would be interesting to show error patterns.

- Only the uncertainty on the bedrock is discussed to justify the differences on the edges of the glacier . Could part of these differences also be due to a too simple parameterization with spatially uniform coefficients and be due to the calibration which uses velocities only along the central line ?

*Minor comments :*

- ***Page 2 line 46 :*** *« the location of the oldest ice in Antarctica »* . Add also reference to Passalacqua et al. (2018) ?
  *Passalacqua O., M. Cavitte, O. Gagliardini, F. Gillet-Chaulet, F. Parrenin, C. Ritz and D. Young, 2018. Brief communication: Candidate sites of 1.5 Myr old ice 37 km southwest of the Dome C summit, East Antarctica, The Cryosphere, 12, 2167-2174, doi:10.5194/tc-12-2167-2018*
- ***Page 3 line 63,*** ref. to Gäggeler et al. Remove *« () »*
- ***Page 4 line 85,*** ref. to GLAMOS. Idem
- ***Page 5 lines 106-107*** *« to iteratively compute the ice flow velocity field and the mass balance »* ***and Sec. 3.1 :*** Clarify that you compute the evolution of the glacier free surface and give the equation.
- ***Sec 3.1.1 Data :***
    - ○ ***Bedrock topography :*** as the uncertainty in the bed is mentioned as a potential explanation for the discrepancy between the model and the data on the sides of the glacier (sec. 5 page 19), it would be interesting to give more details on the method to get the bed and show the radar measurements profiles in Fig.3 ?
    - ○ ***Page 6, lines 114-115 :*** *« and an update of observed velocities based on the 2015-2019 observations »* is that not was is derived from the Sentinel-2 orthoimages mentioned in the previous sentence ?
- ***Figure 3 and similar maps :*** please include a scale in the figures.
- ***Page 6, line 125 :*** change the order of the sentence, i.e. *« ub is the norm of the basal velocity, \sigma and u are the basal shear stresses and basal velocities… ».*
- ***Page 7, Line 128 :*** Provide the exact parametrisation used for the transition between cl and cu.
- ***Sec 3.1.3 :*** give units for fm, ri and rs
- ***Sec 3.1.5 :***
    - ○ justify the fact that you calibrate cl only against 14 points along the central flow line and not against the whole velocity map shown in Fig. 3
    - ○ It would be clearer to explain that you select 3 values for A and 4 values for Cp then calibrate the remaining parameters for the 3*4=12 sets.
- ***Sec 3.2.2 :*** As most of the readership will not be expert in radionuclides it would be interesting to give some description of the values that are expected and the possible interpretation of the different nuclides in this section.
- ***Sec 4.2***
    - ○ ***Line 225 :*** « *By contrast all the samples collected in summer 2018 show very small activities (below 0.1 mBq/kg in 239Pu, results not shown). Therefore, only 2019 sampling results were considered in the following.* ». As I'm not an expert in radionuclides, at this point, it was not clear for me why the 2018 samples show small activities and were disregarded. I understood only later that this is because the 2018 line didn't sampled the right section so that the ice was younger and thus not contaminated.
    - ○ ***Line 244-245 :*** *« We associate it also with a ±2 year uncertainty, which corresponds to the distance between the two peaks. »* We read above that the two maximums correspond to 1958 and 1963 so the distance between the peaks is 5 years and not 2 ?

- *Sec 4.3,*
  - *line 274* : « *Parameters A and cl were found to match observed surface velocities of the ablation area over the 2015-2019 period ».* Please clarify this sentence. In fact cl is calibrated against velocity measurements for each value of A, and it seems that every combination give similar RMSE ?.
  - *lines 276-278* : « *This is due to the new dataset of observed velocities (Appendix A), which showed faster ice in the ablation area than the dataset used formerly. ».* Is it the only reason? As I understand the former study used a unique value for the friction coefficient, so we could imagine that the best value was a compromise between high friction in the upper part and low friction in the lower part ?
  - *Line 280* : « *Thus we explored a stronger mass balance vertical gradient, i.e., higher precipitation and higher melt scenarios, as no direct measurements were available there. »* Not sure why the gradient will be stronger, if you increase precipitation and melt by the same amount the gradient should remain the same ? You explore a larger set of mass balance scenario? Explain the relation with the gradient.
- *Figure 11, caption* : « *Distance traveled by the Dakota main body along the trajectory line drawn on Fig. 10 »* I assume that each parameter set results in a different trajectory, so it is not exactly the one shown in Fig. 10. Maybe, it could be interesting to show the 12 trajectories in Fig. 10 or in a new figure?

---

## Referee Comment (RC2) · Johannes Sutter (Referee) · 21 Jul 2020

Jouvet and colleagues make use of Uranium and Plutonium tracers deposited in the 1950's and 1960's on the Gauligletscher (Switzerland) due to the fallout of nuclear weapons tests. They use these tracers to identify isochrones and benchmark their high resolution glacier model against them. They argue, that using these isochrones as a data benchmark to tune against provides a more stringent constraint on model parameter uncertainties compared to traditionally used tuning targets such as DEMs and surface velocity maps and their results make a compelling point.

The manuscript is well written, with high quality figures and a logical structure which

is easy to follow. Its content fit well into the scope of The Cryosphere as they show convincingly, that their methodology can be used to improve the parameterisation of glacier models significantly.

I congratulate the authors on a really nice manuscript and only have some minor comments/questions followed by stilistic/spelling edits.

-Given the large improvement of the overall model performance due to parameter optimisation a natural question would be how projections of the glacier's evolution would change compared to the old parameter set. Maybe the authors could speculate as to how they think the new parameters might change the expected mass loss in the future or whether they plan to carry out further simulations in this direction.

- To illustrate the effect of the model optimisation an additional figure showing the spatial expression of ice thickness and surface velocity model mismatches with respect to observations (percent, relative change) for the old and new model setting would be nice.

- it is interesting to see in Figure 7, that the parameter optimisation seems to have a moderate effect on RMSE (thickness) compared to the change in observed and modelled glacier length. However, this might be a misinterpretation. As mentioned in the comment above, a 2D figure of the model mismatch for old vs new model would be helpful.

- on page 16 you mention that uncertainties in the bedrock topography could be causing the exaggerated flow asymmetry. This raises the question as to what extent the bedrock topography can be used as a "tuning" parameter to improve the velocity patterns. It probably depends on whether there is a systematic uncertainty in the bedrock topography which would offset the flow asymmetry? In general, modifying the bedrock topography to match flow patterns would be inadvisable unless one knows the actual bedrock topography. But then you would use the corrected bedrock topography in the first place. You re-iterate this point in the discussion by stating that

"Here we have shown that our bedrock is likely too shallow on the north-east side and too deep on the other side"

However, to me it is unclear how you get to that conclusion. The poor model prediction regarding line 1 and 5 might have other causes? Also bedrock uncertainties are to be assumed for the whole glacier area, why should they be especially relevant in case of line 1 and line 5? Please elaborate.

-on page 17 you mention the interesting fact that the model isochrones shift from a U-(after 1970) to a V- shape (before 1970) without giving an explanation. Isn't this just due to the narrowing channel the glacier is pushing through? I would intuitively assume that the glacier moves more homogeneously across a given horizontal transect in the upper part compared to the lower part (where glacier flow is confined on both sides)? A short interpretation as to why the shape changes would be helpful.

minor edits:

p2, l47 I suggest to add

Parennin et al 2017, "Is there 1.5-million-year-old ice near Dome C, Antarctica?" (TC) and Passalacqua et al. 2018, "Brief communication: Candidate sites of 1.5 Myr old ice 37 km southwest of the Dome C summit, East Antarctica" (TC)

p2, l50 combining what to an ice flow model?

p3, l61: ... percolation water remains up to debate.

p3, l74: ... is being expected ...

Figures 3,5,9,10: please include a scale so the reader can appreciate the spacial extent of the glacier.

As a general point I would suggest to introduce the term "contaminated" in the sense that the ice samples are contaminated by U/P, as the casual reader might mistake the term contaminated (e.g. in Figure 5) for an indication that the sample couldn't be used

due to contamination as opposed to the "uncontaminated" samples.

Figure 5: suggest to enlarge the dots in the legend and put "L2" above isochrones and next to "1954".

p7,l 135: ... thick compared to the rest of the glacier? unclear.

p12, l253: unclear what you mean by "associate" here. The 1963 isochrone is associated with a pm 2 year uncertainty?

p15, l292: to a lesser extent

p15, l301: We therefore selected this parameter combination as our best guess set.

Figure 7: for the sake of readability I suggest to make the small markers (discarded runs) semi-transparent. Same fro Figure 8

p15, l305: "displays"

p16, l 306: ... using the original and the revised best-guess model, respectively, and confirms the ...

p16, l311: replace "and then" with "i.e."

p16, l314: to a lesser extend

p17, l325: I suggest to rephrase to: "Ice with radionuclide contamination above 0.25mBq/kg has the same pattern and mostly appears to be band-wise (Fig. 5)"

p17, l329: this corresponds to a mean ?surface? ice flow of ...

p17, l330: suggest to rephrase: However, as the ice was thicker in the past it most likely exhibited faster flow as well.

p18,l1: corresponds to a ...

p18, l345: ...downstream compared to the lateral flowlines and the ... p19,l1: running an ice flow model

p19,l366: Furthermore, snow-covered ...

p19,l377: Here, the bedrock was inferred ...

p19, l380: our bedrock data

p19, l386: ... Alpine glacier. It is important to note, that other tracers ...

p22, l440: suggest to rephrase: In this study, we successfully used Plutonium and Uranium contaminations in an Alpine Glacier induced by the 1950s and 1960s atmospheric nuclear weapon tests to date the ice of the glaciers ablation zone.

p22,l445: The combination

p22, l457: this part is a little implicit. Why is it remarkable? I would assume (correct me if I am wrong) that the climatic conditions of Aletschgletscher changed similarly to the Gauligletscher, so the correction factors should be similar. What are the qualitative differences of Aletschgletscher compared to Gauligletscher (except for the size) which would make a similarity of parameters surprising? I agree with the authors, that isochrones should be utilised to constrain model simulations. However, in this case "traditional" tuning targets for the Aletschgletscher seemed to have produced similar results compared to the fine tuning against isochrones for the Gauligletscher? Or was the 2011 model also optimised with respect to isochrones?

---

## Author Comment (AC1) · 3 Sep 2020

We would like to greatly thank you for your comments on our manuscript.

In this study the authors use measurements of radionuclide activity in ice samples taken from the surface of Gauligletscher to calibrate anice flow model. Radionuclide activity anomalies are typical of the nuclear weapon tests conducted in the 1950s and 1960s and provide a constraint on the ice age.The model is then used to update the

trajectory of an aircraft which crash-landed on the glacier in 1946 and parts of which have recently reappeared. The predicted trajectory appears to be more consistent than that given by a previous study.In addition to the radionuclide data, 2 DEMs and a velocity field are used to constrain the model parameters (3 parameters for ice dynamics (A,cu,cl) and 2 parameters for surface mass balance (Cp and CM)).This is a very original study, the results are convincing and the paper is well presented. I have listed below some general and detailed comments that deserve some considerations by the authors :

**General comments:**

- The parameterization used in this paper for the basal friction differs from the previous study since here 2 distinct coefficients are used, for the upper and lower parts of the glacier. The friction coefficient for the upper glacier is calibrated using the radionuclide measurements and appears to be consistent with what we could guess as it leads to low sliding in the upper part of the glacier. Thus I wonder what is the real contribution of the radionuclide data for the calibration of the model, i.e. is it not possible to calibrate this parameter without these data and/or what would the model give if we assume, for example,no sliding above the equilibrium line?

⇒ The lack of velocity flow data in the accumulation area (snow-covered regions show hardly no features to track in comparison to ablations areas) makes challenging the calibration of the ice dynamics in glacier upper parts. Furthermore, our sliding parametrization is global in space (only two parameters) and in time (no time variation at all), and we have no information at all about former ice flow magnitude. Therefore, we need data that are global in space and time as well for calibration. While our observations are mostly available in the ablation area and in recent times, radionuclide data have the advantage to be global in space and

time, which is a remarkable feature. The only global and past data we have are the DEMs, and even if there are key data, there are not sufficient to constrain our set of 5/6 model parameters. Radionuclide data are especially interesting as they include data from the past while being observable today (which is obviously not the case for DEMs). **The following sentence "In particular, Radionuclide contain data from the past seven decades while being still observable today." was added to the conclusion to emphasize this.** To our knowledge, there is no analogue data that have the same nice property.

⇒ It must be stressed that "fewer sliding in the upper part than in the lower part" is our own assumption based on physical considerations, and is not a result of radionuclide data (see i) in Section 4.3). Our results show that sliding can be reduced in the upper part but not necessarily suppressed. Assuming no sliding at all in the upper part might be too restrictive considering that over-estimated sliding in the upper part could possibly compensate (flux-wise) for underestimated ice deformation in this region as we assumed a constant viscosity parametrization (i.e. constant rate factor $A$).

• The performances of the model are presented only in terms of RMSE, as the parameterizations used are quite simple I think it could be interesting to show some error maps (speed and elevation) in order to discuss the robustness of the model. Notably only the speeds and ages along the centreline are used for the calibration, which corresponds roughly to the aircraft's trajectory. However the ages on the sides of the glacier are less well reproduced by the model and it would be interesting to show error patterns. ⇒ **We added an appendix on "Error pattern between modelling and observations", where we display the error pattern in terms of DEMs for the best guess model after 33 and 63 years of model simulations, and shortly discuss this additional result in the paper.** It is however more difficult to produce a similar map for velocities

as we only considered 14 control points. Lastly, our choice of Figures 5 and 8 was made so that one can visualize partially the observation-modelling misfit in terms of age of ice in both directions: cross-flow (Fig. 5) and flow (Fig. 8).

• Only the uncertainty on the bedrock is discussed to justify the differences on the edges of the glacier. Could part of these differences also be due to a too simple parameterization with spatially uniform coefficients and be due to the calibration which uses velocities only along the central line? ⇒ Thank you, this is a very good point. Indeed, the inaccuracy of the model to reproduce the age of ice along the lateral direction can also be due to an inaccurate parametrization of basal sliding. **We have included this second possible cause along the bedrock data throughout the paper (section 4.3 and fourth paragraph of the discussion).**

**Minor comments:**

• **Page 2 line 46:** "the location of the oldest ice in Antarctica". Add also reference to Passalacqua et al. (2018)? Passalacqua O., M. Cavitte, O. Gagliardini, F. Gillet-Chaulet, F. Parrenin, C. Ritz and D. Young, 2018. Brief communication: Candidate sites of 1.5 Myr old ice 37 km southwest of the Dome C summit, East Antarctica, The Cryosphere, 12, 2167-2174 ⇒ **Done**

• **Page 3 line 63**, ref. to Gäggeler et al. Remove "()" ⇒ **Done**

• **Page 4 line 85**, ref. to GLAMOS. Idem ⇒ **Done**

• **Page 5 lines 106-107** "to iteratively compute the ice flow velocity field and the mass balance" and **Sec. 3.1**: Clarify that you compute the evolution of the glacier

free surface and give the equation. ⇒ **A section on "mass transport" has been added in the model section to provide further details. However, we do not give the equation itself for the safe of conciseness. Instead we refer to an equation in one of the Elmer/Ice core paper.**

- **Sec 3.1.1 Data: Bedrock topography:** as the uncertainty in the bed is mentioned as a potential explanation for the discrepancy between the model and the data on the sides of the glacier(sec. 5 page 19),it would be interesting to give more details on the method to get the bed and show the radar measurements profiles in Fig.3? ⇒ **As mentioned above, we now propose both uncertainties – in the bed and/or on the basal parametrization of sliding – in the revised manuscript.** Details concerning the bedrock derivation from radar profiles and a Figure showing the profiles are given in Supplementary material of our early study by Campagno and al. published in Frontiers in Earth Sciences, 2019, (https://www.frontiersin.org/articles/10.3389/feart.2019.00170/full# supplementary-material). **We added two references to this material in our revised manuscript.**

- **Sec 3.1.1 Data: Page 6, lines 114-115:** "and an update of observed velocities based on the 2015-2019 observations" is that not was is derived from the Sentinel-2 orthoimages mentioned in the previous sentence? ⇒ **"Sentinel-2" was added for clarity.**

- **Figure 3 and similar maps:** please include a scale in the figures. ⇒ The scaling is indicated by a the grid of the topographic map, however, it was not mentioned in caption. **This is now recovered. Concerned figure captions starts read "Topographic map of Gauligletscher with 1 km grid spacing ..."**

- **Page 6, line 125:** change the order of the sentence, i.e. "ub is the norm of the basal velocity ..., $\sigma$ and u are the basal shear stresses and basal velocities... ⇒ **Done**

- **Page 7, Line 128:** Provide the exact parametrisation used for the transition between cl and cu. ⇒ Giving the exact parametrisation is somewhat cumbersome for an unimportant detail. **Instead we added the approximative width of the transition band, which is the main information to retain.**

- **Sec 3.1.3:** give units for fm, ri and rs ⇒ **Done**

- **Sec 3.1.5:** justify the fact that you calibrate cl only against 14 points along the central flow line and not against the whole velocity map shown in Fig.3 ⇒ As we only have one parameter to tune for sliding in the lowest area, our optimization problem would be highly under-determined if we were fitting the entire velocity field. Therefore, we restricted the misfit function (RMSE) to only a few uniformly selected points with the (more modest) goal to get the global velocity magnitude as good as possible. In that perspective, it makes sense to focus on the highest speeds, which are obtained approximatively along the central flowline. **For clarification, the following justification sentence was added: "The choice of considering only 14 points distributed along the central flowline was made to reduce under-fitting in the minimization procedure, which optimizes a single parameter."**

- **Sec 3.1.5:** It would be clearer to explain that you select 3 values for A and 4 values for Cp then calibrate the remaining parameters for the 3*4=12 sets ⇒ We agree that one could have fix other parameters than $A$ and $C_p$, and optimize the remaining ones. Here have 5 parameters, and 3 constrains, so 2 parameters remains free, and we have to make a choice. Any other choice would give other "discretization points" of the parameter space but should yield to the same results. As our choice was more practically-motivated than anything else, we don't see the point to justify it in the text as it has no implications.

- **Sec 3.2.2:** As most of the readership will not be expert in radionuclides it would be interesting to give some description of the values that are expected and the
possible interpretation of the different nuclides in this section ⇒ **We added a paragraph at the beginning of Section 3.2.2 to introduce and motivate this section, as well as to give some expected values from the literature to ease the interpretation for non experts.**

- **Sec 4.2: Line 225:** "By contrast all the samples collected in summer 2018 show very small activities (below 0.1mBq/kg in 239Pu, results not shown). Therefore, only 2019 sampling results were considered in the following.". As I'm not an expert in radionuclides, at this point,it was not clear for me why the 2018 samples show small activities and were disregarded. I understood only later that this is because the 2018 line didn't sampled the right section so that the ice was younger and thus not contaminated. ⇒ **That's a good point. The sentence was completed with "since the ice sampled in 2018 was mostly likely too young for being contaminated (Fig. 5)" for the sake of clarity.**

- **Sec 4.2: Line 244-245:** "We associate it also with a $\pm 2$ year uncertainty, which corresponds to the distance between the two peaks."We read above that the two maximums correspond to 1958 and 1963 so the distance between the peaks is 5 years and not 2? ⇒ Here we mean the interval $[-2.5, 2.5]$ or $\pm 2.5$ has a length of 5 y (1958-1963). We rounded 2.5 to 2.

- **Sec 4.3, line 274:** "Parameters A and cl were found to match observed surface velocities of the ablation area over the 2015-2019 period".Please clarify this sentence. In fact cl is calibrated against velocity measurements for each value of A, and it seems that every combination give similar RMSE?. ⇒ **We rephrased "For each $A$, parameter $c_l$ is ...". Yes, the RMSE is not very different between optimized couples ($A$, $c_p$) so that we can hardly use this constrain/data to exclude parameters.**

- **Sec 4.3, lines 276-278:** "This is due to the new dataset of observed velocities (Appendix A), which showed faster ice in the ablation area than the dataset used

formerly.". Is it the only reason?As I understand the former study used a unique value for the friction coefficient, so we could imagine that the best value was a compromise between high friction in the upper part and low friction in the lower part? ⇒ Yes, this is the main reason. As observed velocities are anyway only available in the ablation area, the misfit function can only be used to assess $c_l$, and not $c_u$. So the former study simply assumed that the tuning made in the ablation area can be extended to the accumulation area. Here we get rid of this assumption and tune $c_u$ using new radionuclide-based data.

- **Sec 4.3, Line 280:** "Thus we explored a stronger mass balance vertical gradient, i.e., higher precipitation and higher melt scenarios, as no direct measurements were available there." Not sure why the gradient will be stronger, if you increase precipitation and melt by the same amount the gradient should remain the same? You explore a larger set of mass balance scenario?Explain the relation with the gradient. ⇒ Here we meant precipitation and melt range or amplitude (controlled by $C_P$ and $C_M$) over the full glacier altitude, making the gradient naturally stronger or weaker, considering that melt is highly controlled by altitude. **For clarity, we changed "higher precipitation and higher melt scenarios" into "higher precipitation and higher melt amplitudes".**

- **Figure 11, caption:** "Distance traveled by the Dakota main body along the trajectory line drawn on Fig. 10"I assume that each parameter set results in a different trajectory, soit is not exactly the one shown in Fig. 10. Maybe,it could be interesting to show the 12 trajectories in Fig. 10or in a new figure? ⇒ When preparing the manuscript, we actually first drew all trajectories, but found that i) it was very difficult to distinguish them ii) it was no very informative. Additionally, Fig. 11 already gives key information on all 12 trajectories, namely the end positions projected on a flowline. Therefore, we did not include an additional figure showing the 12 trajectories in plan-view to not overload this already long paper.

[Figure]

---

## Author Comment (AC2) · 3 Sep 2020

We would like to greatly thank you for your comments on our manuscript.

Jouvet and colleagues make use of Uranium and Plutonium tracers deposited in the 1950's and 1960's on the Gauligletscher (Switzerland) due to the fallout of nuclear weapons tests. They use these tracers to identify isochrones and benchmark their high resolution glacier model against them. They argue, that using these isochrones

as a data benchmark to tune against provides a more stringent constraint on model parameter uncertainties compared to traditionally used tuning targets such as DEMs and surface velocity maps and their results make a compelling point. The manuscript is well written, with high quality figures and a logical structure which is easy to follow. Its content fit well into the scope of The Cryosphere as they show convincingly, that their methodology can be used to improve the parameterisation of glacier models significantly.

I congratulate the authors on a really nice manuscript and only have some minor comments/questions followed by stilistic/spelling edits.

- Given the large improvement of the overall model performance due to parameter optimisation a natural question would be how projections of the glacier's evolution would change compared to the old parameter set. Maybe the authors could speculate as to how they think the new parameters might change the expected mass loss in the future or whether they plan to carry out further simulations in this direction

  ⇒ This is a good point, that we have not discussed in our original manuscript. **We added the following sentence in conclusion about the challenge of well calibrating key parameters for modelled glacier projections: "Yet tuning accurately ice flow and mass balance parameters is essential to reliably model the future evolution of glaciers with a small uncertainty range, especially in the context of global glacier retreat in a changing climate regime."** However, we do not plan any specific simulations of Gauligletscher in the future.

- To illustrate the effect of the model optimisation an additional figure showing the spatial expression of ice thickness and surface velocity model mismatches with respect to observations (percent, relative change) for the old and new

model setting would be nice ⇒ **We added an appendix on "Error pattern between modelling and observations", where we display the error pattern in terms of DEMs for the best guess model after 33 and 63 years of model simulation, and shortly discuss this additional result in the paper.** A similar figure was produced for the former model in the in Supplementary material of the first study by Campagno and al. published in Frontiers in Earth Sciences, 2019, (https://www.frontiersin.org/articles/10.3389/feart.2019.00170/full#supplementary-material). However, one can not see a clear improvement looking at the DEM error patterns between the former and the revised model because DEMs alone are not sufficiently constraining (see also our first block of answer to referee # 1). Say differently, one can obtain a reasonable ice surface elevation at the end of the modelled period while underestimating or overestimating simultaneously internal (ice dynamics) and external (mas balance) ice fluxes, the two compensating each other. Last, we have not produced a similar error pattern map for velocities as we only considered 14 control points, and not the full glacier surface as for DEMs.

• It is interesting to see in Figure 7, that the parameter optimisation seems to have a moderate effect on RMSE (thickness) compared to the change in observed and modelled glacier length. However, this might be a misinterpretation. As mentioned in the comment above, a 2D figure of the model mismatch for old vs new model would be helpful.

⇒ One always needs to be careful with interpreting the fit between observed and modelled glacier lengths since a slight and local change in melt parametrization at the tongue might induce strong change in the glacier length, which is computed along a given central flowline. This explains why the RMSE in terms of glacier length has a wider range that any other RMSEs. For this reason and because there are more global, RMSE in DEMs are – to our opinion – more

reliable metrics to assess model quality.

- on page 16 you mention that uncertainties in the bedrock topography could be causing the exaggerated flow asymmetry. This raises the question as to what extent the bedrock topography can be used as a "tuning" parameter to improve the velocity pat-terns. It probably depends on whether there is a systematic uncertainty in the bedrock topography which would offset the flow asymmetry? In general, modifying the bedrock topography to match flow patterns would be inadvisable unless one knows the actual bedrock topography. But then you would use the corrected bedrock topography in the first place. You re-iterate this point in the discussion by stating that "Here we have shown that our bedrock is likely too shallow on the north-east side and too deep on the other side" However, to me it is unclear how you get to that conclusion. The poor model prediction regarding line 1 and 5 might have other causes? Also bedrock uncertainties are to be assumed for the whole glacier area, why should they be especially relevant in case offline 1 and line 5? Please elaborate

⇒ We agree that optimizing the bedrock is far from obvious. Our statement remains quiet general i.e. one can accelerate (slow down) the flow by making the ice thicker (thinner), however, we are aware that the "how to" is a own field of research, and there is a community using proper inverse modelling dealing with this problem. **To answer your concern we tempered our statement "Here we have shown that our bedrock is likely too shallow on the north-east side and too deep on the other side" into "Here our bedrock data might be too shallow on the north-east side and too deep on the other side to explain the mismatch between modelled and observed isochrones.".** Also, the inaccuracy of the model to reproduce the age of ice along the lateral direction could also be caused by an inaccurate parametrization of the sliding (which is constant in the lower part here) as suggested by the other referee. **Therefore,**

**we have included this second possible cause along the bedrock data throughout the paper (section 4.3 and fourth paragraph of the discussion).**

- on page 17 you mention the interesting fact that the model isochrones shift from a U-(after 1970) to a V- shape (before 1970) without giving an explanation. Isn't this just due to the narrowing channel the glacier is pushing through? I would intuitively assume that the glacier moves more homogeneously across a given horizontal transect in the upper part compared to the lower part (where glacier flow is confined on both sides)?A short interpretation as to why the shape changes would be helpful.

  ⇒ Thank you very much, your interpretation makes perfect sense to us. **We added the following sentence: "Here we can reasonably assume that the downstream tightening of isochrones follows the narrowing of the channel, which hosts the glacier tongue."**

**Minor edits: Most of the suggestions from the referee were followed. Comments with answers from authors are reported below only in the case the suggestions were not or not entirely followed.**

- **Figure 7:** for the sake of readability i suggest to make the small markers (discarded runs) semi-transparent. Same fro Figure 8. ⇒ We tried to make small markers partly transparent, but it did not improve the readability.

- **p17, l325:** I suggest to rephrase to: "Ice with radionuclide contamination above 0.25mBq/kg has the same pattern and mostly appears to be band-wise (Fig.

5)" ⇒ We left the original sentence to keep both radionuclides in the same first introductory sentence before being specific on each.

- **p19,l1:** running an ice flow model

  ⇒ 'tuning' sounds more accurate than 'running' to us.

- **p19,l366:** Furthermore, snow-covered ...

  ⇒ We kept the original formulation as 'Indeed' sounds to us a more appropriate connection word.

- **p19,l1:** p22, l457: this part is a little implicit. Why is it remarkable? I would assume (correctme if I am wrong) that the climatic conditions of Aletschgletscher changed similarly to the Gauligletscher, so the correction factors should be similar. What are the qualitative differences of Aletschgletscher compared to Gauligletscher (except for the size)which would make a similarity of parameters surprising? I agree with the authors, that isochrones should be utilised to constrain model simulations. However, in this case"traditional" tuning targets for the Aletschgletscher seemed to have produced similar results compared to the fine tuning against isochrones for the Gauligletscher? Or was the 2011 model also optimised with respect to isochrones?

  ⇒ Aletsch 2011 model did not involve any observed isochrones for tuning, but instead used traditional data similar as Gauli (DEMs, surface velocities, ...), however, we had more data in past times making the calibration further reliable. Aletsch and Gauli are sufficiently different (especially size-wise) that we do not necessarily expect tuned parameters to be close to each other. As both key models (ice flow) and (mass balance) still rely on simplifications (the most

complex, the most generic), specific re-calibration is required for each glacier, especially if the size differs. **However, we decided to remove noticing the match of the two parametrizations from the conclusion as we can not make a strong point based on a good match of only two glaciers, and it is likely that the range would get larger if we were able to add glaciers to the statistic.**

---

## Author Response (AR1)

September 12, 2020

Dear Dr. De Rydt,

Please find our revised manuscript after taking into account the comments of the two referees, as well as the two minor mistakes you pointed out before the discussion starts. A marked-up manuscript version is attached to this letter. The revisions we made are described in the the answers to the two referees I already posted. We additionally answer specifically in the next two paragraphs the two points that remained rather vague to you.

- In our paper, we emphasize the fact that the radionuclide data contain new valuable information (that other data do not include), and the strength of model constrain is first-of-all driven by the new information available irrespectively of the calibration method. The radionuclide data are much stronger constraints compared to any other ones we used as illustrated by the substantial discrepancy between the former model results (which did not used these data) and the revised ones. **We emphasized this further in the revised manuscript (second paragraph of the conclusion).** To be completely fair, the revised model also took benefit of improved observed velocity data. However, these data i) only concern modern times (no information on the former flow of ice) ii) only concern the lower part of the glacier (no information on the accumulation zone) iii) give no information on the mass balance. By contrast, radionuclide information contains information on long-term, global (ablation and accumulation zones) and mass balance as well. Therefore, this is first-of-all the radionuclide data that permit a strong revision of the model. We are not aware of any other types of data that can perform similarly except any other tracer (or Lagrangian data), whose the availability is rare. About your last question "if other method, such as volume conservation, be used to constrain flow parameters of the upper glacier?" Our calibration relies on the knowledge of several DEMs (which cover the whole glacier and several past time frames). Therefore it also accounts for the mass change/conservation in the accumulation area, however, this is not sufficient to constrain fluxes as a slow glacier with little accumulation and melt can lead to the same surface elevation change than a fast glacier with higher accumulation and melt rates.

- We agree that we missed reviewer 1s comment "It would be clearer to explain that you select 3 values for A and 4 values for Cp then calibrate the remaining parameters for the 3x4=12 sets". **So we added this sentence at the beginning of the section "We selected 3 values $A = 60, 100, 150$ MPa$^{-3}$ a$^{-1}$ and 4 values $C_P = 1, 1.2, 1.35, 1.5$ that were chosen in a physical range. For each pair of parameters ($A, C_P$) among the 12 possible combinations, the procedure to calibrate the three remaining ones ($c_l$, $c_u$, and $C_M$) consisted of several steps (summarized in Fig. 4) which were repeated until the convergence was reached:"**

With my best regards,

On the behalf of my co-authors

Guillaume Jouvet
University of Zurich

[revised manuscript text omitted]